# Towards End-2-End modelling in a consistent NPZD-F modelling framework (ECOSMOE2E_vs1.0): Application to the North Sea and Baltic Sea

Ute Daewel[1], Corinna Schrum[1,2], Jed Macdonald[3,4]

[1]Helmholtz Centre Geesthacht, Institute of Coastal Research, Max-Planck-Str. 1, 21502 Geesthacht, Germany
[2]Geophysical Institute, University of Bergen, Allegaten 41, 5007 Bergen, Norway
[3]Faculty of Life and Environmental Sciences, University of Iceland, 101 Reykjavík, Iceland
[4]Oceanic Fisheries Programme, Pacific Community (SPC), Noumea BP D5 98848, New Caledonia

*Correspondence to*: Ute Daewel (ute.daewel@hzg.de)

**Abstract.** Coupled physical-biological models usually resolve only parts of the trophic food chain and hence, run the risk of neglecting relevant ecosystem processes. Additionally, this imposes a closure term problem at the respective "ends" of the considered trophic levels. Here we aim to understand how the implementation of higher trophic levels in a NPZD model affects the simulated response of the ecosystem, by using a consistent NPZD-Fish modelling approach (ECOSMO E2E) in the combined North Sea and Baltic Sea system. By using this approach we addressed the above-mentioned closure term problem in lower trophic ecosystem modelling at very low computational costs and thus provide an efficient method that requires very few data to obtain spatially and temporally dynamic zooplankton mortality.

On the basis of the coupled ecosystem model ECOSMO II we implemented one functional group that represents fish and one group representing macrobenthos in the 3d model formulation. Both groups are linked to the lower trophic levels and to each other via predator-prey relationships, thus allowing the investigation of both, bottom-up processes and top-down mechanisms in the trophic chain of the North Sea and Baltic Sea ecosystem. Model results for a ten-year long simulation period (1980-1989) were analysed and discussed with respect to the observed patterns. To understand the impact of the newly implemented functional groups for the simulated ecosystem response, we compare the performance of the ECOSMO E2E to that of a respective truncated NPZD model (ECOSMO II) applied to the same time period. Additionally, we performed scenario tests to analyse the new role of the zooplankton mortality closure term in the truncated NPZD and the fish mortality term in the End-to-End model, which summarizes pressure imposed on the system by fisheries and mortality imposed by apex predators.

We found that the model-simulated macrobenthos and fish spatial and seasonal patterns agree well with current system understanding. Considering a dynamic fish component in the ecosystem model resulted in slightly improved model performance with respect to representation of spatial and temporal variations in nutrients, changes in modelled plankton seasonality and nutrient profiles. Model sensitivity scenarios showed that changes in the zooplankton mortality parameter are

transferred up and down the trophic chain with little attenuation of the signal, while major changes in fish mortality and in fish biomass cascade down the food chain.

## 1 Introduction

The majority of spatially-resolved marine ecosystem models are dedicated to a specific part of the marine food web. These models can be differentiated into lower trophic level Nutrient-Phytoplankton-Zooplankton models (so called NPZ models or LTL models) (e.g. Blackford et al., 2004; Daewel and Schrum, 2013; Maar et al., 2011; Schrum et al., 2006; Skogen et al., 2004) and, on the other end of the trophic chain, higher trophic level models (HTL models). The latter mainly simulate fish on a species level including both single-species Individual-Based Models (IBMs, e.g. Daewel et al., 2008; Megrey et al., 2007; Politikos et al., 2018; Vikebø et al., 2007), and multi-species models. Although some of these models are complex and already include many food-web components such as OSMOSE (Shin and Cury, 2004, 2001) or ERSEM (Butenschön et al., 2016), the separation of trophic levels often constrains such models' ability to simulate and distinguish between major control mechanisms on marine ecosystems (Cury and Shannon, 2004). The difficulty to resolve trophic feedback mechanisms increases the uncertainties when modelling the impacts of external controls on the trophic food chain (e.g. Daewel et al., 2014; Peck et al., 2015).

In the last 10 to 15 years, major efforts have been made to link the different trophic levels together to cover the marine ecosystem from the lowest to the uppermost "end" (E2E) (Christensen and Walters, 2004; Fennel, 2009; Fulton, 2010; Heath, 2012; Shin et al., 2010; Travers et al., 2007; Watson et al., 2014). Although, some models such as Atlantis (Fulton et al., 2005), StrathE2E (Heath, 2012) or Ecopath with Ecosim (EwE) (Christensen and Walters, 2004) consider more trophic levels (from phytoplankton to marine mammals and birds and/or fisheries) consistently within the model formulation, the majority of approaches couple conceptually different model types, either 'one-way' with no feedback on the lower trophic levels (e.g. Daewel et al., 2008; Rose et al., 2015; Utne et al., 2012), or 'two-way' (e.g. Megrey et al. 2007, Oguz et al. 2008), when linking the trophic levels. All these approaches work reasonably well in serving a specific purpose or scientific question, but are accompanied by different uncertainties and conceptual limitations to model ecosystem structuring under external forcing. While one-way coupled approaches neglect feedbacks and hence imply difficulties at the model interfaces, comprehensive food-web models like Atlantis resolve food webs on the basis of species or specific groups and are difficult to parameterize especially in complex ecosystems. One of the most commonly used food-web modelling tools is the Ecopath with Ecosim (EwE) modelling software (Christensen and Walters, 2004), which provides an instantaneous snapshot of the trophic mass balance in marine food webs. Together with the dynamical modelling capability (Ecosim) and a tool that replicates the model on a spatial grid (Ecospace) it allows 2d estimates of the systems response to e.g. policy measures. However the approach still falls short in simulating ecosystem dynamics on high temporal and 3d spatial resolution. In Peck et al. (2015), a number of different ecosystem models used in the European project VECTORS (http://www.marine-vectors.eu/) were reviewed. Besides discussing statistical and physiology-based life cycle models, Peck et al. (2015)

identified strengths and weaknesses in food-web models like Atlantis. While the strength of these models is the explicit consideration of species-specific responses, which are often vital for informing advice for management, a clear weakness of those models is the huge amount of data needed for model parameterization (Peck et al., 2015) and the sensitivity to assumptions made on the food-web structure and functioning. Another drawback of the recent End-2-End models is the lack

in spatial resolution. They are either solved in 2d as EwE or, as Atlantis is, resolved in predefined (based on environmental conditions) larger area polygons. This consequently excludes the dynamic resolution of ecologically highly-relevant hydrographical structures such as tidal fronts or the thermocline, and implies that future changes in relevant hydrodynamics and their impacts cannot be considered through these models. To our knowledge, the first approach that attempted to resolve the trophic food web more consistently in a functional group framework with the spatial and temporal resolution of a state of

the art physical model is the food web model presented by Fennel (2010, 2008) and Radtke et al. (2013). They proposed a nutrient-to-fish model where fish is included consistently in a NPZD model framework using a Eulerian approach. In their study they chose a species-specific way to introduce fish using size-structured formulations for the three major fish species in the Baltic Sea. The model has been proven to work well in the Baltic Sea, which is characterized by a relatively simple food web (Casini et al., 2009; Fennel, 2008), but is likely more difficult to parameterize in other, more complex structured

food webs involving more key species such as the North Sea.

Here, we aim to address these conceptual limitations in End-to-End modelling and present a different approach based on the assumption that food availability and corresponding energy and mass fluxes are the major controls in higher trophic production and spatial and temporal distribution of the fish biomass. A physical-biological coupled 3d NPZD ecosystem model is extended to include fish and macrobenthos (MB). Although our idea is inspired by the model presented by Fennel

(2010, 2008) and Radtke et al. (2013), it is substantially different to their concept based on three key species. We used instead a functional group approach, which represents the entire fish population, and aims to be consistent with the functional group approach used for phytoplankton and zooplankton. This enables estimation of the total fish production potential and allows for resolving structuring impacts on the ecosystem. The advantage of this generic approach is its broad applicability. It allows for general and comparative studies on changing ecosystem structure and is not limited by unknown

changes in key species for the respective ecosystems. The approach we use cannot address changes in ecosystem structure related to variations in the fish assemblage or selected fishing activities. It does, however, provide the potential for further developments towards a more complex food web (e.g. by distributing fish into separate feeding guilds), which will then allow us to address specific changes in food web structure.

We will present a first application of the Eulerian End-to-End model for the shelf sea system of the North Sea and Baltic

Sea. The coupled North Sea and Baltic Sea system (Figure 1) is located adjacent to the North Atlantic Ocean. Despite their close proximity, they are very different in physical and biogeochemical characteristics. The North Sea features pronounced co-oscillating tides combined with a major inflow from the North Atlantic. The Baltic Sea in contrast, has only a narrow opening to the North Sea leading to an almost enclosed, brackish system with weak tidal forcing (Müller-Navarra and Lange, 2004). The restricted exchange capacities and fresh water excess to the Baltic Sea lead to an estuarine-type circulation with

strong stratification and relatively low salinities and to a relatively long water residence time of about 30 years (Omstedt and Hansson, 2006; Rodhe et al., 2006). Due to its brackish waters, winter sea ice develops regularly in the Baltic Sea, which might, in severe winters cover almost the entire surface (Seinä and Palosuo, 1996).

The two systems also differ substantially in terms of ecosystem dynamics. The North Sea is known as a highly productive area inhabited by more than 26 zooplankton taxa (Colebrook et al., 1984) and over 200 fish species (Daan et al., 1990), with highest biomasses distributed among demersal gadoids, flatfish, clupeids and sandeel (*Ammodytes marinus*) (Daan et al., 1990). Consequently, the North Sea is economically highly relevant with nine nations fishing in the area with landings of currently about 2 million tons annually (ICES, 2018b). Compared to the North Sea, species composition in the Baltic Sea is primarily limited by the low salinities and encompasses only a few key players for zooplankton (Möllmann et al., 2000) and fish (Fennel, 2010). Thus, compared to the North Sea, commercial fishing in the Baltic Sea includes only a few stocks with total landings of over 0.6 million tons annually (ICES, 2018a). In both regions, landings peaked in the 1970s and have substantially (ca 50%) declined since then. Thus, fishing has a substantial impact on the overall fish biomass in the region.

Studies on the food web dynamics of the North Sea and Baltic Sea highlight additionally the relevance of benthic fauna for fish consumption (Greenstreet et al., 1997; Tomczak et al., 2012). The term benthos refers generally to all organisms inhabiting the sea floor. A comprehensive review on the topic has been given by Kröncke and Bergfeld (2003). The faunal components encompass over 5000 species generally divided by size into microfauna, meiofauna, and macrofauna. Additional differentiation can be made under consideration of the vertical habitat structure, with infauna inhabiting the inner part of the sediment and epifauna living above the sediments. While in the North Sea the macrobenthos assemblages are structured based on the spatial distribution of sediment characteristics and depth, the Baltic sea community is additionally influenced by oxygen availability (Ekeroth et al., 2016) and salinity (Gogina et al., 2010). Besides its role as prey and predator in the marine foodweb, marcobenthos additionally influences nutrient effluxes from the sediments and thus can modify temporal and spatial patterns in nutrient concentrations (Ekeroth et al., 2016).

Here we present a functional-type, E2E modelling approach, which relates food availability to potential fish growth and biomass distributions. In this manuscript we introduce the conceptual basis of the model, discuss its characteristics and explore its performance with respect to observed fish and MB distributions. Further, we analyse model performance at the lower trophic levels in comparison to the NPZD modelling approach, and discuss the potential of our model to understand and compare basic regional ecosystem characteristics.

## 2 Methods

### 2.1 Model Description

The E2E model builds on the coupled hydrodynamic-lower trophic level ecosystem model ECOSMO II (Barthel et al., 2012; Daewel and Schrum, 2013; Schrum et al., 2006; Schrum and Backhaus, 1999), which is further expanded for the present

study. The latter model has been shown to accurately reproduce lower trophic level ecosystem dynamics in the coupled North Sea and Baltic Sea system. The model equations and a model validation on the basis of nutrients were presented in detail by Daewel & Schrum (2013), who showed that the model is able to reasonably simulate ecosystem productivity in the North Sea and Baltic Sea on seasonal up to decadal time scales. The NPZD module was designed to simulate different

macronutrient limitation processes in targeted ecosystems and comprises 16 state variables. Besides the three relevant nutrient cycles (nitrogen, phosphorus and silica), three functional groups of primary producers (diatoms, flagellates & cyanobacteria) and two zooplankton (herbivore & omnivore) groups were resolved. Additionally, oxygen, biogenic opal, detritus and dissolved organic matter were considered. Sediment is implemented in the model as an integrated surface sediment layer, which accounts for consideration of sedimentation as well as resuspension. Biogeochemical remineralisation

is considered in surface sediments leading to inorganic nutrient fluxes into the overlying water column. To allow for nutrient specific processes in the sediment, the organic silicate content of the sediment is estimated in as separate state variable. A third sediment compartment is considered for iron-bound phosphorus in the sediment (Neumann and Schernewski, 2008). To estimate total fish production and biomass in a consistent manner compared to lower trophic level production, we expanded the NPZD-type model by implementation of a wider food web in the system (Figure 2).

Although zooplankton in ECOSMO II could in principle grow also at the bottom, its parameterization as a passive tracer and the choice of parameterization for the functional groups makes it unsuitable for representing benthic production. Hence, the parameterization of a specific functional group representing benthic (meio- and macro-) fauna remains necessary. The group was designed similar to the zooplankton groups, but with the additional restriction that benthos grows only at the bottom and is neither exposed to advection nor diffusion. Benthic fauna has been observed to exhibit little tolerance to hypoxic and

anoxic conditions (Kröncke and Bergfeld, 2003), wherefore macrobenthos growth was estimated only for positive oxygen concentrations in the model framework. In contrast to the benthic compartment in ERSEM (Butenschön et al., 2016), where the benthic predators are distributed into three different functional types, we here neglect different functional traits of infauna and epifauna and consider only one functional group, which we, for convenience, will refer to as macrobenthos (MB).

Each state variable C in ECOSMO II is estimated following prognostic equations in the form

$$C_t + (\vec{V} \cdot \nabla)C + (w_d)C_z = (A_v C_z)_z + R_c \tag{1}$$

with $C_t = \frac{\partial c}{\partial t}$, $C_z = \frac{\partial c}{\partial z}$ where t is time and z is the vertical coordinate. The equation includes advective transport $(\vec{V} \cdot \nabla)C$ ($\vec{V} = (u, v, w)$ current velocity vector), vertical turbulent sub-scale diffusion $(A_v C_z)_z$ ($A_v$: turbulent sub-scale diffusion coefficient), sinking rates $(w_d)C_z$ ($w_d$: sinking rate is non-zero only for detritus, opal and cyanobacteria)) and chemical and biological reactions $R_c$. Since MB production occurs locally at the bottom and the group is not exposed to mechanical

displacement, Eq. 1 is simplified for MB to

$$\frac{dC_{MB}}{dt} = [R_{MB}]_{z=z_{bottom}} \tag{2}$$

Concurrently chemical and biological interactions are employed in the biological reaction term $R_C$, which is different for each variable ($C$) based on the relevant biochemical processes. For MB it divides into production, which is a function of consumption ($R_{MB\_Cons}$) and assimilation efficiency ($\gamma_{MB}$) and a MB loss term.

$$R_{MB} = \gamma_{MB}R_{MB\_Cons} - R_{MB\_Loss} \tag{3}$$

The production of zooplankton in the model depends on the available food resources, which include phytoplankton, detritus, and, for the omnivorous zooplankton group, also herbivorous zooplankton. For the macrobenthos functional group, we assume a much wider range of potential prey items. The benthic community can be divided into benthic suspension/filter feeders feeding mainly on phytoplankton, detritus and bacteria, benthic deposit feeders ingesting bottom sediments, and larger individuals exerting predation pressure (among others) on the available zooplankton (Kröncke and Bergfeld, 2003).

Thus, the prey spectrum of the simulated MB functional group includes, besides phyto- and zooplankton, also detritus and organic sediments. Since we assume that benthic suspension/filter feeders would also indirectly ingest dissolved organic matter, we chose to add the latter to the MB diet.

Consumption of the MB group is estimated as the sum of consumption rates for the single prey items (herbivorous zooplankton ($Z_1$), omnivorous zooplankton ($Z_2$), flagellates ($P_1$), diatoms ($P_2$), detritus (DET), dissolved organic matter

(DOM), organic sediment (SED1))

$$R_{MB\_Cons} = C_{MB}\left(\sum_{j=1}^{2} G_{MB}\left(C_{Z_j}\right) + \sum_{l=1}^{2} G_{MB}(C_{P_l}) + G_{MB}(C_{DET}) + G_{MB}(C_{DOM}) + G_{MB}(C_{SED1})\right) \tag{4}$$

Grazing rates $G_{MB}$ on prey type $X$ ($X \epsilon [Z_1; Z_2; P_1; P_2; DET; DOM; SED1]$) are estimated using the Michaelis-Menten equation (Michaelis and Menten, 1913; Monod, 1942):

$$G_{MB}(C_X) = \sigma_{MB,X}\frac{a_{MB,X}C_X}{r_{MB}+F_{MB}} \tag{5}$$

where $F_{MB} = \sum_X a_{MB,X}C_X$ .

The half saturation constant $r_{MB}$ and values for grazing rates ($\sigma_{MB,X}$) are given in Table 1 and feeding preferences ($a_{MB,X}$) are given in Table 2.

The MB loss term consists of excretion ($\varepsilon_{MB}C_{MB}$), natural mortality ($m_{MB}C_{MB}$) and predation mortality from the fish functional group ($C_{Fi}G_{Fi}(C_{MB})$). Values for excretion ($\varepsilon_{MB}$) and mortality rate ($m_{MB}$) are given in table 1.

$$R_{MB\_Loss} = C_{Fi}G_{Fi}(C_{MB}) + m_{MB}C_{MB} + \varepsilon_{MB}C_{MB} \tag{6}$$

We assume that fish is a prognostic variable, which is, in contrast to the other prognostic state variables in the ecosystem model, not exposed to passive transport processes and not moving actively horizontally. This can be translated into the assumption that characteristic fish migration is restricted to a spatial scale below the model grid size in the order of 10km.

Larger scale migration behaviour is neglected here. Since we know that neglecting larger horizontal migrations places major constraints on the model's ability to estimate the spatial distribution of the overall fish biomass, we think that the assumption is still valid for calculating the overall fish production potential and its spatial distribution in the system. In the following, we

will thus refer to "fish" as a functional group that comprises the fish biomass that emerges based on the lower trophic production at each horizontal grid cell. For clarification it needs to be noted that, even when called "fish production potential", the fish biomass is a state variable in the model that interacts dynamically with the lower trophic level components and that will be used in the following to confirm the models ability to simulate spatial and temporal patterns of carbon transfer to higher trophic levels. On the other hand by constraining the horizontal migration capabilities of the fish group to one grid cell we will likely underestimate the local fish production potential by confining it to the locally available fish biomass.

The potential "fish" still needs to be considered more mobile than the other ecosystem components in the model, wherefore the vertical distribution of the fish group is assumed to result from fish active movement and varies based on food availability. This leads to the following principles to be applied for the fish functional group:

1. We neglect horizontal fish migration larger than the spatial scales of one grid cell.

2. Fish is mobile and, within the given time step (20 min), able to search the water column for food beyond the vertical extent of a single grid cell. Therefore, we assume that fish is able to utilize the food resources available in all depth levels of the water column. Consequently fish is not, as all the other variables, calculated within one grid cell only, but depends on the vertically-integrated food resources.

3. The vertical distribution of the fish group and fish production depends on the food availability in each grid cell. That means that during each time step the integrated fish biomass in the water column is vertically redistributed based on the vertical prey distribution after consumption was estimated.

Following the three principles implies that equation 1 is simplified to $\frac{\partial C_{Fi}}{\partial t} + w_m(z)\frac{\partial C_{Fi}}{\partial z} = R_{Fi}$, where $w_m(z)$ is the vertical migration speed, which is given implicitly by the vertical distribution of the fish biomass in dependence of the vertical prey distribution. In each grid cell the biological interaction term ($R_{Fi}$) is estimated containing fish consumption ($R_{Fi_{Cons}}$), assimilation efficiency $\gamma_{Fi}$ and a loss term ($R_{Fi_{Loss}}$).

$$R_{Fi} = \gamma_{Fi}R_{Fi_{Cons}} - R_{Fi_{Loss}} \tag{7}$$

Following principle 2 and 3, $R_{Fi}$ is estimated with a two-step process. First, total fish consumption at the horizontal location (m,n) is estimated based on the vertically integrated values for fish and prey biomass.

$$R_{Fi_{prod}}(m,n) = \sum_{k=1}^{k_{max}}(R_{Fi_{Cons}}(m,n,k) * \Delta z_k) = \sum_X(G_{Fi}(P_X) * P_{Fi}) \tag{8}$$

Where $P_X$ ($X$ is one of four prey types (herbivorous zooplankton $Z_1$, omnivorous zooplankton $Z_2$, detritus DET, macrobenthos MB) available for fish in the model) is defined as the integrated biomass of the prey type $X$ ($P_X = \sum_{k=1}^{k_{max}}(C_{X_k} * \Delta z_k)$ ) over all vertical levels (k=1:$k_{max}$) at the respective horizontal location (m,n). $P_{Fi}$ is the corresponding vertically integrated

biomass of fish. Grazing rates $G_{Fi}$ are estimated using the Michaelis-Menten equation $G_{Fi}(P_X) = \sigma_{Fi,X}\frac{a_{Fi,X}P_{Fi}}{r+F}$ with $F = \sum_X a_{Fi,X}P_X$ and $a_{Fi,X}$ are the feeding preferences of fish on prey type $X$ (values in Table 2), in a similar manner as for the zooplankton and MB groups.

In a second step, fish consumption in each grid box (m,n,k) is estimated by weighting the prey specific components of the consumption in each vertical layer based on the vertical distribution of the prey biomass with $C_X(m,n,k)/P_X(m,n)$ such that

$$R_{Fi_{Cons}}(m,n,k) = P_{Fi}\left(\sum_{j=1}^{2} G_{Fi}\left(P_{Z_j}\right) * \frac{C_{Z_j}}{P_{Z_j}} + G_{Fi}(P_{DET}) * \frac{C_{DET}}{P_{DET}} + [G_{Fi}(P_{MB})]_{k=bottom}\right) \tag{9}$$

Note that, since fish do not tolerate anoxic conditions, only grid cells featuring positive oxygen concentrations were considered for the estimate of fish consumption.

The loss term for fish includes mortality and excretion.

$$R_{Fi_{Loss}} = m_{Fi}C_{Fi} + \varepsilon_{Fi}C_{Fi} \tag{10}$$

Mortality is considered as a linear mortality rate including biomass losses due to natural mortality and predation. Fisheries mortality has not been considered for the standard simulation, but was explicitly addressed in additional scenario experiments as described in section 2.4. Excretion is considered to be related to fish metabolism and consequently to respiration (Table 4 equation for oxygen) and hence has been parameterized in dependence of temperature (Clarke and Johnston, 1999; Gillooly et al., 2001). Reaction kinetics vary with temperature according to the Blotzmann's factor k and we formulated the fish excretion as

$$\varepsilon_{Fi} = \mu_{Fi}e^{\left(\frac{\theta_{Fi}}{k}*TK\right)}; TK = \frac{T-T_0}{T*T_0} \text{ with T is given in °K and } T_0=273.15 \text{ °K.} \tag{11}$$

All rates are given in Table 3.

Fish and macrobenthos predation, excretion and mortality are considered in addition to the pelagic lower trophic level biological reaction terms (see Daewel and Schrum, 2013) for nutrients, phytoplankton, zooplankton, detritus, dissolved organic matter or sediment (Table 4). While fecal matter is accounted for through the use of assimilation efficiency, the excretion term from both fish and MB contributes directly to the nutrient reaction terms (see Table 4 equation for phosphate and ammonia). The new zooplankton mortality term consists of fish predation and additional background mortality, which is 80% of the background mortality term used in ECOSMO II. In-situ and laboratory studies indicate that predation mortality accounts for 67-75% of the total mortality (Hirst and Kiørboe, 2002). Other sources of mortality are parasitism, disease and starvation. Including fish and macrobenthos as predators in the model does not, however, account for the overall predation exerted on zooplankton. By analyzing the pelagic food web of the North Sea Heath (2005) identified fish consumption of omnivorous zooplankton being on average 6.7 gC m$^{-2}$ year$^{-1}$. That value was recalculated in Heath (2007), after more specifically considering the role of fish pre-recruits feeding on zooplankton, to amount to ~7.6 gC m$^{-2}$ year$^{-1}$, while the average consumption by carnivorous zooplankton (euphausids and macroplankton) is considerably higher with 11 gC m$^{-2}$ year$^{-1}$. Since the zooplankton groups in the model are not stage resolving, intraguild predation is not explicitly prescribed as

a mortality term, but is implicitly included in the background mortality. Although our model results also suggest that a substantial amount of zooplankton is consumed by macrobenthos in the shallow regions of the North Sea, assuming that about 20-30% of zooplankton mortality stems from the combined fish and benthos group seems to be a good first guess. Please consider that the reduction of the background mortality rate to 80% of its initial value does not necessary imply that the background mortality is 80% of the total mortality. By including a spatially and temporally variable mortality term in the model, this term can locally play a much larger (or smaller) role for the overall mortality. To evaluate the model sensitivity to the choice of this parameter, we performed scenario experiments described in section 2.4. The degradation products from MB, fish mortality and food consumption contribute to particulate (POM) and dissolved organic matter (DOM). The latter is distributed between the two partitions POM and DOM with a ratio 60%/40% (for explanation see Daewel and Schrum 2013b). Since MB is living at the sea floor we assume that the generated POM is directly contributing to the sediment pool (SED1), which via re-suspension might contribute to the suspended particulate matter (DET) in dependence of bottom stress. The fish contribution to the particulate organic matter is added to the detritus pool.

## 2.2 Experimental Setup

To evaluate the model performance after including two new functional groups, we chose to analyse model results from a 10-year long simulation period (1980-1989) based on two key requirements. First, since the characteristic time scale of the Baltic Sea is in the range of three decades and also the model results indicate an adaptation period of about 20-30 years for fish and MB in the Baltic Sea (not shown), the simulation was started in 1948 to allow for a sufficiently long spinup period with a realistic forcing. Second, we wanted to analyse a relatively undisturbed period with respect to hydrodynamic and biogeochemical conditions. Therefore we chose a period prior to the observed regime shift at the end of the 1980's and prior to the major Baltic inflow in 1993.

The model setup is similar to the one described by Daewel and Schrum (2013) for the long term simulations with a hydrodynamic core model based on HAMSOM (Hamburg Shelf Ocean Model) as described in (Schrum and Backhaus, 1999) with additional modification of the advection scheme (Barthel et al., 2012). The model is formulated on a staggered Arakawa-C grid with a horizontal resolution of 6' x 10' (~10km) and a 20-min time step. The vertical dimension was resolved with 20 vertical levels, whereof the upper 40 m have a layer thickness of 5 m and the resolution is coarsening below that. The model requires boundary conditions at the atmosphere-ocean boundary (NCEP/NCAR re-analysis (Kalnay et al., 1996)) and at the open boundaries to the North Atlantic. Transport of freshwater and nutrient loads from land is considered. Details on the utilized boundary and forcing data are given by Daewel and Schrum (2013), who also gave a detailed description of analysis methods and validation datasets.

## 2.3 Datasets and statistical methods for model analysis

As described in Daewel and Schrum (2013), we used observational data on surface (depth <10m) nutrients (nitrate and phosphate) in the North Sea, which are made available by ICES (International Council for the Exploration of the Seas,

www.ices.dk), for nutrient validation. Observations and modelled surface nutrients were averaged over the upper 10 m of the water column and co-located in space and in time and corresponding statistics were calculated for the subareas specified in Figure 1. The seasonal cycle has not been removed prior to the analysis. The reason for doing so is the sparse data situation, which does not allow estimating a reliable seasonal cycle at each location. Additionally, the seasonal cycle changes from

year to year. Thus, removing an average seasonal cycle from the data would add a bias to the data and hence increase the level of uncertainty. In the Baltic Sea, we used vertical nutrient profiles ($NO_3$, $PO_4$, $O_2$) at two distinct locations in the Baltic proper (BY5 (Lat/Lon: ~55.1°N/15.59°E), BY15 (Lat/Lon: ~57.2°N/20.03°E) see Figure 1) from the Baltic Sea monitoring network (see e.g. www.helcom.fi) that were continuously sampled since 1970. The data are available for download at www.ices.dk (accessed 05/2012). To account for inconsistencies in sampling frequencies, we co-located model data and

observations prior to estimating average vertical profiles and standard deviations. For this purpose the model values were linearly interpolated onto a 1m vertical grid to allow best local comparison to the observations, while the observations where considered on the actual sampling depth. The statistical measures chosen for model analysis are the Pearson's correlation coefficient, the standard deviation and the root mean square deviation presented in a Taylor diagram (Taylor, 2001), and Empirical Orthogonal Functions (EOFs) as described in Storch and Zwiers (1999). The EOF analysis is a statistical method

to understand major modes of variability in multidimensional data fields. A detailed description on how this method has been applied is given in (Daewel et al., 2015): "The annual values of the spatially explicit variable field form a NxM matrix $\chi$ (N: number of years; M: number of wet grid points). The empirical modes are given by the K eigenvectors of the covariance matrix with non-zero eigenvalues. Those modes are temporally constant and have the spatially variable pattern $p_k(m=1,…,M)$ where $k=1,…,K$. The time evolution $A_k(t=1,…,N)$ of each mode can then be obtained by projecting $p_k(m)$

onto the original data field $\chi$ such that $\chi(t,m) = \sum_{k=1}^{K} p_k(m)A_k(t)$. In the following we will refer to $A_k(t)$ as the principal components (PC) and to $p_k(m)$ as empirical orthogonal function (EOF). The percentage of the variance of the field $\chi$ explained by mode k is determined by the respective eigenvalues and is referred to as the global explained variance $\eta_g(k)$. Before using the method to analyse the spatiotemporal dynamics of the field, the data were demeaned (to account for the variability only) and normalized (to allow an analysis of the variability independent of its amplitude). The identified modes

are not necessarily equally significant in all grid points of the data field. Thus, the local explained variance $\eta_{local,k}(m)$ could provide additional information about the regional relevance of an EOF mode and the corresponding PC in percent:

$$\eta_{local}{}^k(m) = \left[1 - \frac{Var\left(\chi(m,t)-p^k(m)A^k(t)\right)}{Var(\chi(m,t))}\right] \cdot 100 \quad , \tag{12}$$

where $Var(X) = \sum_{t=1}^{N}\left(\overline{X} - X(t)\right)^2$ denotes the variance of the field X(t)."

Information on the North Sea fish community are collected in the North Sea international bottom trawl survey (NS-IBTS) (ICES, 2012) and are freely available at the International council for the exploration of the sea (www.ices.dk). The NS-IBTS

dataset contains spatially-resolved, species-specific information on fish length (for some target species also age) and catch per unit effort (CPUE: in numbers captured per hour). Given that our model estimated state variables on the base of carbon biomass, we converted fish length and abundance data to fish biomass (in grams captured per hour) based on published length-weight relationships (LWRs) for each species sampled in the NS-IBTS survey between 1980-1989 inclusive.

LWRs were derived from Coull et al. (1989), Froese et al. (2014) and FishBase (Froese and Pauly, 2000) for teleost species, McCully et al. (2012) and Templeman (1987) for rays and skates, Klaoudatos et al. (2013) for crabs, and Pierce et al. (1994) and Guerra and Rocha (1994) for squids. Total body weight (W) is a function of total length (L) following the relationship $W = aL^b$ where $b$ is a parameter indicating isometric growth in body proportions if $b \sim 3$, and $a$ is a parameter describing body

10   shape and condition if $b \sim 3$.

Calculations were made at the species level, where species names were available, with the $a$ and $b$ parameter estimates taken from published sources in the following order:-

1. Studies specific to the North Sea (i.e. Coull et al., 1989; Froese and Pauly, 2000). If not available then:-

2. Studies specific to the British Isles (i.e. Coull et al., 1989; Froese and Pauly, 2000; McCully et al., 2012). If not

15       available then:-

3. Studies specific to the North Atlantic (i.e. Coull et al., 1989; Froese and Pauly, 2000; McCully et al., 2012; Templeman, 1987). If not available then:-

4. Posterior mean estimates of $a$ and $b$ from a Bayesian hierarchical analysis for the species across all credible LWR studies, regardless of location (see Froese et al., 2014; Froese and Pauly, 2000).

For some species, e.g. herring, sprat, mackerel, Coull et al. (1989) provided mean parameter estimates for each month, or at least some months throughout the year, in addition to annual estimates. In these cases, we used annual estimates only. If information on genus only, family only or order only was available, the $a$ and $b$ parameter estimates for all species within that genus, family or order captured during the surveys were averaged to give a genus-, family- or order-specific value. If no

25   information was available at any species level within a genus, family or order, then the geometric mean of that genus, family or order was computed from the data file accompanying Froese et al. (2014). The data were assorted onto the horizontal model grid based on their sampling location. For the data-model comparison we will only consider data from the first quarter of the year, since in the considered time period (1980-1989) this was the only systematically surveyed quarter in the survey.

## 2.4 Scenario Definition

30   Two sets of scenarios were performed to evaluate the simulated food-web response to specific changes in the food web parameterizations. We study the structuring effects of the new model closure term (higher trophic levels/fisheries) and the effects of changes in background zooplankton mortality. The first set of scenario experiments addresses the carbon loss through apex predators and fisheries by adding another source of mortality in addition to the natural fish mortality term.

Following the fisheries overview of the North Sea and Baltic Sea region as published by ICES (2018a, 2018b), biomass losses due to fisheries are in both regions in the range of 20%-50% of the total fish biomass per year. For convenience we therefore decided to receive the fisheries mortality rate by scaling the natural mortality rate (=0.001d$^{-1}$ =0.365yr$^{-1}$) with 0.5, 1 & 2. As such, three different scenarios were calculated with an average (0.1825 yr$^{-1}$), high (0.365 yr$^{-1}$) and extreme (0.5475 yr$^{-1}$) loss rate. Note that the latter two loss rates were chosen to provoke extreme responses in the fish biomass, and do not represent realistic catch rates in the areas.

The second set of scenario experiments was designed to understand the ecosystem response to changes in the zooplankton natural mortality. This term previously formed the sole closure term of the system and the rate has been reduced by 20% to account for the additional mortality induced to the system by MB and fish predation. Here, we chose four scenarios for the experiment. First we defined the control run, which considers the unchanged zooplankton mortality rate from ECOSMO II. Then, control -20%, is the reference setup for ECOSMO E2E. Additionally, we discussed control -40% and control +20% for comparison.

## 3 Results & Discussion

In the following, we discuss the basic characteristics of the model and assess its performance based on 10-year averages of the model variables. Specifically, we i) present and discuss the spatial dynamics of the newly introduced functional groups; ii) we discuss the seasonality of the ecosystem components and introduce the MB and fish diet composition emerging from the model; iii) we present the comparison of the simulated fish biomass distribution to observed data and repeat the nutrient validation analysis as previously presented for ECOSMO II in Daewel and Schrum (2013); and iv) we discuss the model sensitivity with respect to ecosystem model closure.

### 3.1 Description of modelled spatial pattern

The mean spatial patterns of calculated MB and fish vertically integrated biomass for the period (1980-1989) are presented in Figure 3. On average, estimated MB biomass (Figure 3a) in the North Sea is 1.98 gC m$^{-2}$ for the considered time period. As we will see later, this value is highly sensitive to the parametrization of zooplankton mortality and fisheries effort (c.f. 3.4 and Figure 12). Heip et al. (1992) proposed an average of 7g ash-free dry weight (AFDW) m$^{-2}$ based on a synoptic sampling of North Sea benthos in April-May 1986. This equates to approximately 3.5 gC m$^{-2}$ when assuming a carbon fraction of ash-free dry weight of 0.5 (Ingrid Krönke, Senckenberg am Meer, Wilhelmshaven, Germany, pers. communication), which is somewhat higher than our model estimates, but also includes benthic carnivore biomass (Greenstreet, 1997). Greenstreet (1997) estimated the biomass of the benthic filter feeder and deposit-feeder guild to amount to ~3 gC m$^{-2}$ when the same carbon fraction of 0.5 is applied. Note that the comparison can only be an approximation due to the high variability in carbon content among species (e.g. Timmermann et al., 2012). Observational estimates of North Sea MB biomass indicate a

decrease in biomass with increasing latitude according to Heip et al. (1992), and similar results were obtained from a subsequent sampling project in 2000 (Rees et al., 2007). Particularly high values of MB biomass were found in the shallow areas of the southern North Sea, including the coastal areas and Dogger Bank (Heip et al. 1992 their Figure 1), and in the river mouth areas along the English coast.  This is in clear agreement with what has been estimated by our model.

In the Baltic Sea, the MB biomass was modelled to be on average 1.01 gC m$^{-2}$. This is in the range of what has been published by Timmermann et al. (2012) based on HELCOM data, who reported spatially-resolved values between 5-100 gWWt m$^{-2}$ (approx. 0.25 – 5 gC m$^{-2}$), as well as by Tomczak et al. (2012) who estimated macrobenthos biomass of about 30 t km$^{-2}$, which equals 1.5gC m$^{-2}$ using an Ecopath with Ecosim Baltic Proper food web model. The spatial distribution of MB modelled with our simplified model is consistent with the spatial distribution of major MB species in the Baltic Sea as

presented by Gogina and Zettler (2010) based on species specific model estimates and observations. This applies specifically to the high abundances in the southern Baltic Sea, the near coastal areas and the Gulf of Riga.

Our model estimates highest MB biomass in both North Sea and Baltic Sea in shallower areas, especially near the coast and in bank regions, such as Dogger Bank, Fisher Banks and Oyster Ground, with a maximum of around 5 gCm$^{-2}$ to be found in the southern North Sea. MB production in the model is constrained by the availability of oxygen; therefore, large areas of the

central Baltic Sea are not inhabited by macrobenthos. In the North Sea, minimum MB biomass is estimated slightly offshore of the British coast and in the deeper parts of the Norwegian Trench region. Those minima in the North Sea were not caused by anoxic conditions, but by a lack of prey in the respective areas. The transition zone between North Sea and Baltic Sea, including the Skagerrak, Kattegat, Danish straits and the Fehmarn Belt, exhibits, in contrast, generally high values of MB biomass.

Simulated spatial variability in vertically integrated fish biomass shows a structured pattern both in the North Sea and the Baltic Sea. In the Baltic Sea, maxima of fish biomass are simulated in the coastal areas, the Gulf of Riga, in the southern Baltic Sea including Arkona Basin, Bornholm Basin and Bay of Gdansk and in the Åland Sea at the entrance to the Bothnian Sea. The deeper parts of the Eastern Gotland Basin, the Gulf of Finland and the Gulf of Bothnia in contrast feature very low fish biomass due to low prey biomass and oxygen depletion near the bottom. The modelled spatial distribution compares

well with findings from the nutrient-to-fish model from Radtke et al. (2013). They integrated the model over a 4-year period 1980-1983. In their model approach fish follows specific rules for horizontal migration (food availability, spawning). However, their simulated spatial distribution of the combined biomass for the 3 different simulated fish species is very similar to our estimates. Differences between the simulated fish distributions occur specifically in time periods when predefined spawning areas determine the distribution. Other spatial differences were simulated for the Gulf of Finland where

the model by Radtke et al. (2013) estimated relatively high fish biomasses in contrast to our model, and around Gotland where our model produces fish biomass maxima potentially fostered by the additional availability of macrobenthos as prey, which remains unconsidered by Radtke et al. (2013). Interestingly, the distribution of fish biomass maxima, estimated by our model, resembles the pattern described as cod nursery areas in the Baltic Sea by Bagge et al. (1994).

The structure of modelled North Sea fish biomass is very distinct with maxima in frontal areas such as the tidal mixing front in the southern North Sea and around Dogger Bank and the frontal zone off the German, Danish and British coast, and in the Norwegian Trench. Maxima are also modelled in the Fisher Banks and Oyster Ground and Fladen Ground regions. Minima, in contrast, were estimated in the deeper parts of the western North Sea off the British coast, the German Bight and in the English Channel. They partly resemble the minima estimated for MB, which indicates that fish biomass minima are caused by food shortage in areas with low MB biomass. Especially off the British Coast, studies from Callaway et al. (2002) and Jennings et al. (1999) indicate high fishing effort, indicating that the model likely underestimates the fish biomass in that region. Potential reasons are the model underestimating zooplankton production in that area (Daewel et al., 2015), and the missing impact from the open boundary, where neither zooplankton nor fish were prescribed to enter the model domain.

When integrating fish biomass over the North Sea and Baltic Sea regions, it amounts to ca. 0.462 million tonsC and 0.312 million tonsC respectively. Assuming the carbon content of fish being 45% (Huang et al., 2012; Sterner and George, 2000) and the AFDW (ash free dry weight) to wet weight fish ration ranging from 0.1 to 0.2, this corresponds to a simulated total fish biomass in the range of 5.13-10.27 million tons for the North Sea and 3.47-6.93 million tons for the Baltic Sea. Since the AFDW to wet weight ratio is highly variable, even within species, depending on e.g. temperature, season and diet of the fish (Elliott and Hemingway, 2002), it is difficult to determine an exact value for the simulated entire fish assemblage biomass. The modelled estimates of fish biomass are well within the range of what has been estimated for total fish biomass based on observations for the Baltic Sea (Thurow, 1997). Using yield data and age composition data in catches, Thurow (1997) estimated total fish biomass in the Baltic Sea for the time period 1900-1985. His results indicate relatively low fish biomass (<2 million tons) for the first half of the century, but a drastic increase thereafter. For the time period considered here (i.e. 1980-1989) he proposed the fish biomass to be around 7 million tons. Following ICES (2018b, 2018a), fisheries during the 1980's were in the range of 0.7-1 million tons in the Baltic Sea and 2-3 million tons in North Sea. Despite that the model underestimates fish production due to the no horizontal migration assumption and no fish migration over the lateral boundaries, the model's estimates of fish biomass in the North Sea would support the fisheries landing during that time period.

Estimates for North Sea total biomass for the time 1983-1985 based on the ICES International young fish survey (IYFS) and the English ground fish survey (EGFS) were published by Sparholt (1990). For the first quarter Sparholt (1990) estimated an average fish biomass of about 8.6 million tons, while for the third quarter the average biomass was estimated to be 13.1 million tons. The discrepancy between first and third quarter was explained by the migration of the western stock of Atlantic mackerel (*Scomber scombrus*) and horse mackerel (*Trachurus trachurus*) into the North Sea, which is not considered by our model. Further our results find agreement with output from an Ecopath with Ecosim food-web model of the North Sea as proposed by Mackinson and Daskalov (2007), which estimated that the North Sea total fish biomass was ~11 million tons in the year 1991. Indeed, our modelled estimate is well within the range of previously published observations and model results. We suggest that any discrepancies are most likely due to our model neglecting fish migration at large scales. When discussing the spatial variation of estimated fish biomass we need to consider that the model is constrained by the

assumption that fish do not move horizontally. Thus, we estimate the production potential for fish in each horizontal grid cell rather than the actual fish biomass at a given time and location.

## 3.2 Seasonal dynamics of ecosystem components and diet composition

Values for both MB and fish biomass vary in the course of the year (Figure 4). While the modelled seasonal amplitude for

fish biomass is relatively small (North Sea: 94.7 mgC m$^{-3}$≅11%; Baltic Sea: 84.8 mgC m$^{-3}$≅9.7% of the mean biomass) when

compared to the average values, MB seasonality is substantial (North Sea: 4.4 gC m$^{-3}$≅222%; Baltic Sea: 1.86 gC m$^{-3}$≅183% of the mean biomass). Minimum MB and fish biomass is estimated for winter and early spring and the seasonal maximum is modelled for late summer and autumn. The MB maximum lags behind the zooplankton maximum by about 3 months. In contrast to zooplankton, the MB minimum does not reach values close to zero but the model simulates a significant standing

stock for MB also during winter.

In Figure 4 the seasonal cycles for the phytoplankton and zooplankton estimates of the ECOSMO E2E run is presented together with those of the ECOSMO II simulation (Daewel and Schrum, 2013). The seasonal cycles for both phytoplankton and zooplankton are clearly impacted by the consideration of MB and fish. Although the general phytoplankton biomass seasonality and the phenology remain relatively unchanged, the magnitude of the seasonal maximum, especially the diatom

bloom, is significantly increased in spring and early summer in both regions NS and BS when the MB/fish groups are included. The consideration of seasonally-variable MB and fish predation on zooplankton imposes a different seasonality on zooplankton mortality compared to the constant mortality rate used in ECOSMO II (Daewel & Schrum, 2013) and hence impacts zooplankton phenology. The reduced zooplankton biomass in the beginning of the season due to MB and fish predation (Figure 5) consequently leads to a reduction in phytoplankton mortality and to an increase in phytoplankton

biomass (top-down process). Additionally, MB competes with zooplankton for resources and thereby changes zooplankton seasonality, especially in autumn in the North Sea when MB biomass is highest and it preys dominantly on dead organic material and phytoplankton (Figure 5).

An overview on the seasonal feeding dynamics can be obtained by identifying the monthly prey composition for MB (Figure 5a) and fish (Figure 5b) in the North Sea and Baltic Sea. For MB, the major food source throughout the year is organic

sediments followed by dead organic material, while the percentage of the latter is considerably higher in the North Sea than in the Baltic Sea, presumably due to the fact that a higher percentage of detritus is re-suspended in the tidally influenced, highly turbulent areas of the North Sea. Zooplankton and phytoplankton are included in the diet when available in spring and summer. The fish prey composition (Figure 5b) is very similar in both sub-areas; with MB dominating the diet in the autumn and winter months and omnivorous (large) zooplankton dominating in summer. Detritus contributes a significant food source

in March and April, while small zooplankton appears in autumn only in very low amounts. Greenstreet et al., (1997) reviewed food web studies in the North Sea and analyzed the food consumption of fish by guild. When adding up the average MB and zooplankton in the diet of the four in the study fish guilds considered (demersal piscivores, demersal

benthivores, pelagic piscivores, pelagic planktivores), the ratio zooplankton/MB in the diet lies at around 6/4 in summer, which is comparable to our estimated food composition in summer. In contrast to our model results, the estimates from Greenstreet et al. (1997) show no significant seasonal variations in diet composition. Explanations for this disagreement might be found in the model performance of e.g. the zooplankton standing stock. The latter has been estimated to be very low in winter, and hence lead to an intensification of the modelled zooplankton seasonal cycle and to too little zooplankton in the fish diet in winter. Another possible reason for the mismatch between the model and the estimates from Greenstreet et al. (1997) might be related to spatiotemporal differences in the fish biomass and diet.

An EOF analysis on the monthly mean fields for MB and fish biomass reveals the spatial-seasonal pattern. In Figure 6 (MB) and 7 (fish) the first two EOF patterns are shown for MB and fish biomass respectively. Additionally the local explained variance and the related temporal pattern (PC) is given. For MB, the seasonal signal is very homogeneous across the whole area (Figure 6). With 77%, the first mode explains a significantly large part of the overall variability and the temporal signal resembles the average variability shown in Figure 4. This highlights that the MB seasonality is mainly induced by the seasonal pattern of the system productivity with increased production of fresh organic material in summer and less food availability in winter. This is in line with observations on seasonality of benthic infauna at three different locations in the North Sea published by Reiss and Krönke (2005), who found maximum biomass in late summer. Although the observed seasonality showed the highest magnitude in the German Bight the seasonality was clear at all three locations. The authors concluded that of the potential relevant factors (food availability/quality, water temperature, predation, hydrodynamic stress) food quality plays the major role for infauna seasonality, thus is strongly related to primary production. They also suggest food limitation and predation pressure to be the main processes for decrease in abundance during winter. The same authors also looked at seasonality in the epibenthic community (Reiss and Krönke, 2004) showing that the epifaunal biomass varies less seasonally, especially in the off shore region, and that the main processes causing seasonal variations are related to migratory behavior, which is not covered by our model. For the Baltic Sea only very local studies in seasonality of MB are available of which some indicate locally strong seasonality (Anders and Möller, 1983), while in other regions no seasonal changes in biomass were observed due to the dominance of long-lived species (Persson, 1983). In general the comparison to observations indicates that on the one hand the model is able to represent the main seasonality in MB even though epi- and infauna are not separated. On the other hand, in future studies the consideration of an additional functional group encompassing longer-lived species will be required for addressing MB seasonality more correctly.

The second EOF explains about 16% of the overall variability and is especially important in the Gulf of Finland and the Bothnian Bay where it explains to up to 80% of the total variability. $PC_2$ differs from $PC_1$ by showing a maximum in MB in late autumn and winter with a time lag of about two months compared to $PC_1$, while the minimum is modelled for July and August. The ecosystem seasonality in the Bothnian Bay and the Gulf of Finland is highly impacted by a relatively long period of winter sea ice cover and the onset of the spring bloom is therefore delayed (see e.g. Andersson et al., 1996; Daewel and Schrum, 2013). This would consequently affect the phenology of MB and fish in that area and explains the difference in the seasonal cycle.

In contrast to the MB pattern, the EOF of the fish biomass seasonality (Figure 7) reveals a clear distinction between seasonal signals in different regions. Together, the first two EOFs explain over 70% of the overall variability whereas the first EOF comprises 44%. This first pattern describes a seasonal cycle with a minimum in March/April and a plateau in maximum fish production between August and December, which dominates the average seasonal cycle shown in Figure 4. It particularly

explains the seasonality in the deeper central North Sea, the Norwegian Trench, the Skagerrak and the northern Kattegat region as well as the coastal areas of the central Baltic basins. In all of these regions this mode explains up to 80 % of the variability. The second pattern describes the seasonal variability in the shallow areas of the southern North Sea, including Dogger Bank, the North-Western North Sea and the Belts at the entrance to the Baltic Sea, with a maximum in fish biomass in late spring and summer. The dynamics in these areas are determined by the zooplankton seasonality featuring, unlike the

central North Sea, a maximum in summer. The two modes of the estimated fish seasonal cycles clearly indicate two different fish habitats, structured by food availability and temperature. In Figure 8 fish production, partitioned into diet components, in the shallow (Figure 8a) and deeper (Figure 8b) North Sea is illustrated together with the seasonal temperature cycle. The main differences contributing to differences in fish biomass seasonality are the timing of the food resources and the difference in temperature. In the shallow areas of the North Sea, zooplankton forms the major food source for fish in early

spring and summer, reaching a maximum in May and June. After that, the MB contribution increases and resumes the role as the major food source in August. In the deeper parts of the NS, the dynamics in fish diet composition are shifted by 1-2 months, and, in contrast to the shallower NS, dead organic material plays an important role throughout the year. Since the seasonality in total fish production is very similar in both regions, this difference in diet composition would per se not lead to a difference in fish biomass seasonality as seen from the EOF analysis (Figure 7). However, in addition to the difference in

food resources the two habitats feature very different seasonal temperature cycles. The most likely explanation for the stronger decrease in fish biomass in shallow NS in August and September (Figure 7 $PC_2$) is the temperature driven higher loss rate (Eq. 11).

Here, we can identify the distinction of NS fish communities approximately at the 50m depth line, which is comparable to the separation line reported for North Sea fish communities in earlier published observational studies (Callaway et al., 2002;

Rees et al., 1999). Using data from 270 stations distributed over the whole North Sea, Callaway et al. (2002) separated the NS fish community into several clusters (3 or 5 in dependence of the trawling method) and two main groups. The most conspicuous boundary was defined approximately at the 50m depth contour separating the community in the shallow southern North Sea, which mainly consists of small non-commercial species, from the community in the central North Sea, which was dominated by haddock (*Melanogrammus aeglefinus*), *M. merlangus*, herring (*Clupea harengus*), and plaice

(*Pleuronectes platessa*). The authors also suggested the environmental conditions in the region to play a major role in structuring the community. Although our model cannot distinguish between different species and actual communities, the results indicate a clear distinction between the seasonality and the driving environmental conditions of fish production potential in the shallow southern North Sea and in the central North Sea.

Spatial variations of biomass specific mortality related to zooplankton consumption by fish and MB are given in Figure 9 as an average from 1980-1989. The results were additionally separated into 1st and 2nd quarter of the year to identify potential intra-annual variations as suggested by Maar et al. (2014). The results show a very distinct spatial pattern for fish induced zooplankton mortality (Figure 9a) with increased values in some specific regions in the central North Sea, especially in the

vicinity of Dogger Bank, close to the English coast, Oyster ground and in the Fisher Bank (Little and Great Fisher Bank) area. Our model shows furthermore considerable consumption in the Norwegian Trench, along the coast of the southern and central Baltic Sea including the Kattegat/Skagerrak region, in the central basins of the southern Baltic Proper, the Gulf of Gdansk and in the Gulf of Riga. The biomass specific mortality related to MB consumption (Figure 9b), in contrast, is confined to shallow areas with a relatively strong coupling between the benthic and pelagic system. This includes the

shallower areas of the southern and central North Sea and the near coastal areas in the Baltic Sea. The difference between the 1$^{st}$ and 2$^{nd}$ half-year is relatively small for the fish induced mortality. However, there is a clear but small increase in the central North Sea and a much stronger change for the Baltic Sea pattern with a higher impact in the second half of the year. Also for MB the impact is substantially stronger for the 2$^{nd}$ half-year, clearly related to the strong seasonal signal in MB biomass.

While we estimate zooplankton predation losses within the model, an earlier study by Maar et al. (2014) proposed spatial-temporal variations in biomass specific mortality of zooplankton based on data of the major zooplanktivorous fish species and on larval distribution in the North Sea (their Figure 10C,D). Our results show clear similarities in magnitude and spatial structure when compared to the results from Maar et al. (2014). The results of Maar et al. (2014) however showed a clear difference between 1$^{st}$ half-year and 2$^{nd}$ half-year with decreased biomass specific mortality in the 2$^{nd}$ half-year in the central

North Sea. This intra-annual variation is not evident in our model results. However we found a clear difference in magnitude when comparing winter and summer season (not shown). The reasons for the discrepancies between our model results and Maar et al (2014) are presumably related to interannual variations in fish consumption, which are not considered in a 10-year average, the fact that migration is not considered in the model and thus restrict the spatial variation, and that our functional group cannot resolve species- and stage-specific spatial and temporal variations like e.g. the increase in larval biomass in

spring and changes in species composition. On the other hand, the approach from Maar et al. (2014) reveals uncertainties due to the fact that only parts of the North Sea fish assemblage is considered and that the fish biomass is prescribed and not dynamically coupled to zooplankton biomass.

The approach from Maar et al. (2014) provides the possibility to replace the spatially- and temporally-invariant closure term usually used in NPZD type models (Daewel et al., 2014) by a data driven, detailed formulation, and allows for consideration

of the predation effects of different fish species and larvae on the zooplankton dynamics. However, the main disadvantage is that, as the authors already pointed out, a huge amount of detailed species specific and potentially undersampled data is required and that the estimated mortality index relies on a number of assumptions concerning e.g. the relevance of the individual fish species and spawning time and distribution. Moreover, such a data driven approach has a limited potential for future projections and sensitivity studies on various effects on the ecosystem. However, following a very different, less

detailed approach, the spatial variability of our estimates of zooplankton consumption (Figure 9a) by fish compares surprisingly well with the spatial variability of the fish consumption index provided by Maar et al. (2014; their Figure 4), which we consider an implicit validation of our model approach. The consideration of MB-related zooplankton predation mortality is an additional advantage that arises from our modelling approach.

## 3.3 Model performance and nutrient dynamics

To get a more direct measure for the validity of the modelled fish functional group we used data from the NS-IBTS. In Figure 10, we compared the mean fish functional group biomass distribution to the fish biomass from the NS-IBTS as calculated following the method described in section 2.3. We classified species within the NS fish community into 'demersal' (Figure 10a) and 'pelagic' groups (Figure 10b) based on life-history characteristics, then summed the biomass of each group to form a 'combined' (Figure 10c) group. In contrast to the species-specific differentiation into groups used for the observations, the model results do not provide this level of detail. Here, the differentiation has been performed based on the vertical distribution of the fish biomass. Thus we assigned all biomass in the bottom layer to the 'demersal' groups and biomass in the remaining water column to the 'pelagic' group. Therefore and because the units in the NS-IBTS data and in the model data differ the Figures are not quantitatively comparable. When we compare the data for the different fish groups we find the 'demersal' fish more strongly increasing with latitude, and, following the North Sea bathymetry, with depth. The 'pelagic' fish group biomass, in contrast, shows, in addition to the increase with latitude, a maximum in the south and in the north of the North Sea. Altogether, we find a clear increase in fish biomass with latitude with a maximum at the entrance to the North Sea, but also higher values in the central North Sea and around Dogger Bank. From the data, we can also conclude that the contribution of pelagic fish biomass to the overall biomass is higher in the south than further north. From a pure qualitative comparison, we find the modelled fish group biomass roughly resembling the observed pattern, with increasing biomass from south to north. The model also represents the maximum in the Kattegat and at the northern shelf edge. However, the model estimates a pronounced minimum off the British coast, which is not evident from the observations and likely stems from the zero boundary condition, meaning that no fish enters or leaves the model area over the lateral boundaries, and missing migration parameterization in the model. In summary, we found that the model is able to represent the spatial fish distribution in the North Sea. However, the differences in fish biomass off the British and partly at the European continental coast and the discrepancy between the observed and modelled 'pelagic' group indicate a potential underrepresentation of the pelagic fish stock by the simulated fish functional group.

To understand the effect of changes in the NPZD model closure on model performance with respect to nutrient dynamics, we repeated the nutrient validation for surface nutrients in the North Sea (Figure 11) and for nutrient profiles in the Baltic Sea (Figure 12) as described by Daewel and Schrum (2013). For both surface nitrate (Figure 11a) and phosphate (Figure 11b) the statistics, presented here in a Taylor diagram, indicate an improvement for some regions when MB and fish were considered. Larger improvements occur in regions with relatively high estimated biomass for MB and fish, such as in region E off the English coast, where the correlation coefficient for nitrate improved from under 0.4 to 0.5, and for phosphate from under 0.6

to above 0.7, yet with a stronger bias for both nutrients. Better results were also accomplished in the central North Sea (region K), where the standard deviation moved significantly closer to that of the observations. Small improvements are also shown for regions F and L.

The MB and fish group potentially alters the nutrient dynamics in the Baltic Sea (Figure 12). Although we found only relatively small changes for phosphate relative to the ECOSMO II simulation in both considered locations, clear differences for the nitrogen and oxygen profiles are apparent, especially in the intermediate depth levels between 50 m and 150 m and at the surface. The model indicates a slight upward shift of the oxycline when fish and MB are resolved, which also affects nitrate by relocating the nitrate maximum. This results in decreased model performance with respect to nitrate in the intermediate layer, but improves the performance at the surface by increasing the initially too low modelled surface nutrient concentrations. Ammonium is significantly improved in lower layers at BY15 station for the ECOSMO-E2E model. Several processes interact to determine the changes in vertical nutrient profiles. Possible candidates are, e.g., changes in the oxygen dynamics by including oxygen dependent fish and MB, changes in sediment dynamics, since organic sediments are ingested by MB and subsequently nutrients are released on different time and spatial scales. Additionally, we found that the nutrient dynamics are also sensitive to the parameter choice of zooplankton mortality and the loss rate though fisheries and apex predation (cf. 3.4).

### 3.4 Ecosystem response to structuring drivers

With the two sets of scenarios we try to evaluate the impact of changes in model closure (fisheries mortality) and zooplankton mortality on ecosystem structure.

In the fisheries-scenarios we would expect a top-down response of the ecosystem dynamics to changes in fisheries, such that reduced fish biomass would relax the predation on the secondary producers (zooplankton and MB), which consequently would increase in biomass and reduce phytoplankton (see e.g. Cury et al. (2003)). Our model results indicate this type of trophic response for the Baltic Sea ecosystem (Figure 13b), but with very little efficiency for the lowest trophic level. The reduction of fish biomass for the highest catch rate scenario is about 98% compared to the control run, and for zooplankton and MB the increase is 13% and 62% respectively. The response of phytoplankton biomass, in contrast, is only in the order of 4% and hence small compared to the interannual variability of phytoplankton biomass, which is in the order of 10%.

The North Sea ecosystem responds less predictably than the Baltic Sea to simulated changes in the fish model closure term. Although the reduction of fish biomass in the North Sea results in an increase in MB, zooplankton biomass does not respond correspondingly (Figure 13a). The introduction of a moderate loss term leads to a comparably strong reduction of zooplankton biomass, while, with further increase in the loss rate zooplankton biomass increases again. As in the Baltic Sea, phytoplankton biomass is reduced with increasing fishing effort but the response is even smaller (~3%). The most likely reason for the more complex response of the North Sea ecosystem is the tighter coupling between MB and zooplankton and phytoplankton (see Figure 7a). Since zooplankton forms a prey group for MB in the North Sea, a major change in MB and

fish biomass affects the relevance of the two zooplankton predator groups and the increased predation pressure by MB will counteract (and potentially overrule) the relaxed predation by fish.

The second set of scenario experiments was designed to understand the ecosystem response to changes in the zooplankton natural mortality. In the new E2E model configuration a change of this term cascades up and down the trophic food chain
(Fig 13c&d). In both systems, a reduction in zooplankton natural mortality leads consequently to an increase in zooplankton biomass and to a decrease in phytoplankton and an associated decrease in MB. The difference between the systems becomes manifest in the response of the fish group, which is positive in the Baltic Sea (Figure 13d), but reverses in the North Sea (Figure 13c) with higher fish biomass in a low zooplankton environment. This response highlights again the major role of the MB in the North Sea ecosystem, which partly competes with zooplankton and forms a major prey item for fish.

Despite the strong changes in the magnitude of phytoplankton and zooplankton biomass, the phenology of the seasonal cycles was almost not impacted by the sensitivity changes (not shown). The only distinct change is a decrease in phytoplankton spring biomass in the Baltic Sea when the model closure term is increased. Almost none other of the phenological changes described in section 3.2 was affected when fish biomass was decreased in the first set of sensitivity experiments, highlighting the dominant role of MB for these changes.

**4 Conclusion**

We presented here a 3d resolved food web model that is based on a functional group approach ranging from nutrients to fish. Differently to the study by Fennel (2010), we did not distinguish between different fish species to avoid uncertainties associated with the choice of fish species and their contribution, compared to the unconsidered remaining biomass. Our approach integrates the full production potential for fish into one single functional group by defining the feeding pathways
via primary and secondary production, including zooplankton and MB. This has certain advantages. For example, we avoid the parameterization of a detailed species dependent food web, and moreover, the adaptability of the model to other ecosystems is independent of the local fish assemblage. The advantage of the generic functional group approach used in the model for all trophic levels is that we can simplify a complex community structure and reduce the information to the basic common features, thus avoiding a huge parameter set and data requirements. Still the model is able to simulate relevant
ecosystem dynamics at high spatial and temporal resolutions with relatively low computational requirements.

Despite the simplicity of the approach, we found the model able to reproduce the observed spatial pattern and magnitude of both macrobenthos and fish biomass in the North Sea and Baltic Sea as described in the literature (see Figure 3 and section 3.1). This highlights the advantage of our approach, and adds weight to the assumption that fish biomass distribution
consequently emerges from prey availability and environmental conditions. Furthermore, the model was able to distinguish between the two different fish production areas, separated around the 50-m depth line (compare Callaway et al., 2002), with differences in the seasonal cycle and the diet composition. Although this differentiation is not based on species composition

as in Callaway et al. (2002), it shows that the basic concept that biotic and abiotic conditions determine the composition of the local fish community, as realized in the model, and allows conclusions to be formed about the local fish community, even when it is not explicitly prescribed in the model. This opens up possibilities for additional investigations on, e.g., the inter-annual variability of fish production and biomass through general fish diet composition, and how this compares to observed long-term fish stock variations. Future model developments and applications should particularly address the composition of local fish communities, by classifying fish in two or more functional groups such as planktivores and piscivores, or into pelagic and benthic feeding guilds, to allow for a clearer representation of the food web structure.

However, the simplicity of the model and the related assumptions confine the model interpretation, and some of the simplifications require a revision in future model applications. Besides the redistribution of the MB and fish into several food-web specific functional groups, neglecting fish movements in the model approach is a clear limitation, since we know that fish are mobile and would migrate in response to e.g. food shortage, spawning behaviour or predators. In contrast to the Norwegian Sea, where distinct feeding migrations are observed for the pelagic fish component (Nøttestad et al., 2011) following the northward progressing zooplankton blooms and light conditions, the North Sea and Baltic Sea exhibit a relatively constant spatial pattern of system productivity, with highly productive areas along the coast and less productivity in the central seasonally-stratified regions. Hence, the migratory movements of North Sea and Baltic Sea fish stocks might not be based solely on large feeding migrations, but may also related to temperature and salinity changes and spawning behaviour (Hinrichsen et al., 2016; Hunter et al., 2003; Pinto et al., 2018; Radtke et al., 2013). Additionally, fish migrate into the area from the North Atlantic (e.g. Sparholt, 1990). Two questions arise specifically related to this topic: i) Is including migration strategies on a functional group level effective and reasonable in the North Sea and Baltic Sea environment, considering the variability among species? (ii) Would migration behaviour effectively impact the productivity of the system at the higher trophic level? We would therefore like to highlight the necessity to investigate the impact of specific migration strategies in continuative studies.

One major aim of the model development was to solve the closure term problem that arises with NPZD type models when choosing a fixed zooplankton mortality term (for review see Daewel et al. (2014)). The model results show that the inclusion of a higher trophic functional group can provide a more consistent and dynamic closure term, which produces a realistic but variable mortality field independent of in situ observations, in contrast to observational-based zooplankton predation as used e.g. by Maar et al. (2014). In return, the closure term problem is transferred to the new "end" of the food web, namely the fish group mortality. In future studies, this should be addressed by including dynamic formulations for apex predators and fisheries. This might be accomplished by introducing simple fisheries catch rates as explored in the scenario runs in this study (section 3.4), or by coupling the model to socio-economic models, which allows inclusion of social interests and management decisions in the modelling approach (e.g. Charles, 1989; Schlüter et al., 2014). The latter approach could allow the model to be applied in a fisheries management context – i.e. if the model's ability to capture local fish community structure with respect to potential production and species composition is further developed.

*Code and data availability*. Model code access and data can be obtained upon request. The code is available in the Helmholtz-Centre-Geesthacht git repository https://coastgit.hzg.de/udaewel/hamsom-ecosmoe2e/, and licensed under apache license version 2.

*Author contributions* UD and CS designed the study and developed the modelling approach. UD performed the simulation and data analysis. JM contributed to the study by compiling a comprehensive dataset of fish biomass from the IBTS data. All co-authors contributed to the writing of the paper.

## 5 Acknowledgements

This work is a contribution to the FP7-SeasERA SEAMAN Collaborative Project financed by the Norwegian Research
Council (NRC-227779/E40). We would like to thank Marie Maar for her constructive comments on an earlier version of the manuscript.  Furthermore, we are grateful to an anonymous reviewer and Hagen Radtke, whose thoughtful comments helped a lot to improve our manuscript.

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

| Abbr. | Definition | Value | Units |
|---|---|---|---|
| $r_{MB}$ | MB half saturation constant | 0.5 | mmol C m$^{-3}$ |
| $m_{MB}$ | MB mortality rate | 0.001 | day$^{-1}$ |
| $\varepsilon_{MB}$ | MB excretion rate | 0.025 | day$^{-1}$ |
| $\gamma_{MB}$ | Assimilation efficiency | 0.75 | |
| $\sigma_{MB,X}$ | Grazing rate | 0.1 | day$^{-1}$ |

**Table 1. Parameters used for the MB functional group reaction terms**

| Y \ X | P | Z1 | Z2 | DET | DOM | SED1 | MB |
|---|---|---|---|---|---|---|---|
| MB | 0.2 | 0.2 | 0.3 | 0.1 | 0.1 | 0.1 | - |
| Fi | - | 0.25 | 0.45 | 0.05 | - | - | 0.25 |

**Table 2: Feeding preferences for macrobenthos (MB) and fish (FI) $^{a_{Y,X}}$**

| Abbr. | Definition | Value | Units |
|---|---|---|---|
| $r_{Fi}$ | Fish half saturation constant | 0.7 | mmol C m$^{-3}$ |
| $r_{Fi,MB}$ | Fish half saturation constant (MB prey) | 0.9 | mmol C m$^{-3}$ |
| $m_{Fi}$ | Fish mortality rate | 0.001 | day$^{-1}$ |
| $\mu_{Fi}$ | Fish excretion rate | 0.002 | day$^{-1}$ |
| $\theta_{Fi}$ | T control parameter excretion | 0.5 | |
| $\gamma_{Fi}$ | Assimilation efficiency | 0.7 | |
| $\sigma_{Fi,X}$ | Grazing rates F on MB, $Z_{1,2}$ | 0.01 | day$^{-1}$ |
| $\sigma_{Fi,D}$ | Grazing rates F on D | 0.005 | day$^{-1}$ |
| k | Boltzmann's factor | 8.6173324*10$^{-5}$ | eV K$^{-1}$ |

**Table 3. Parameters used for the fish functional group**

| State variable | Reaction term |
|---|---|
| Phytoplankton | $R_{P_{1,2}} = R_{P_{1,2}} - \left[C_{MB}G_{MB}(C_{P_{1,2}})\right]_{n=bottom}$ |
| Zooplankton | $R_{Z_{1,2}} = R_{Z_{1,2}} - G_{Fi}(P_{Z_{1,2}}) * \dfrac{C_{Z_{1,2}}}{P_{Z_{1,2}}} - [C_{MB}G_{MB}(C_X)/\Delta z]_{n=bottom}$ |
| Detritus | $R_{DET} = R_{DET} - P_{Fi}G_{FI}(P_{DET}) * \dfrac{C_{DET}}{P_{DET}} - [C_{MB}G_{MB}(C_{DET})]_{n=bottom}$ $\qquad\qquad + 0.6((1 - \gamma_{Fi})R_{Fi_{Cons}} + m_{Fi}C_{Fi})$ |
| Dissolved organic matter | $R_{DOM} = R_{DOM} - [C_{MB}G_{MB}(C_{DOM})]_{n=bottom} + 0.4$ $\qquad\qquad * \left(((1 - \gamma_{Fi})R_{Fi_{Cons}} + m_{Fi}C_{Fi}) + ((1 - \gamma_{MB})R_{MB_{Cons}}\right.$ $\qquad\qquad \left. + m_{MB}C_{MB})\right)$ |
| Sediments | $R_{SED1} = R_{SED1} - C_{MB}G_{MB}(C_{SED1}) + 0.6 * ((1 - \gamma_{MB})R_{MB_{Cons}} + m_{MB}C_{MB})$ |
| Phosphate/Ammonia | $R_{PO_4/NH_4} = R_{PO_4/NH_4} + \varepsilon_{Fi}C_{Fi} + [\mu_{MB}C_{MB}/\Delta z]_{n=bottom}$ |
| Oxygen | $R_{O_2} = R_{O_2} - c_{C:O2}[\varepsilon_{Fi}C_{Fi} + [\mu_{MB}C_{MB}/\Delta z]_{n=bottom}]$ $\quad c_{C:O2}$:conversion factor |

**Table 4. Changes in the biogeochemical reaction terms (R) of ECOSMO due to the macrobenthos (MB) and fish functional groups.**

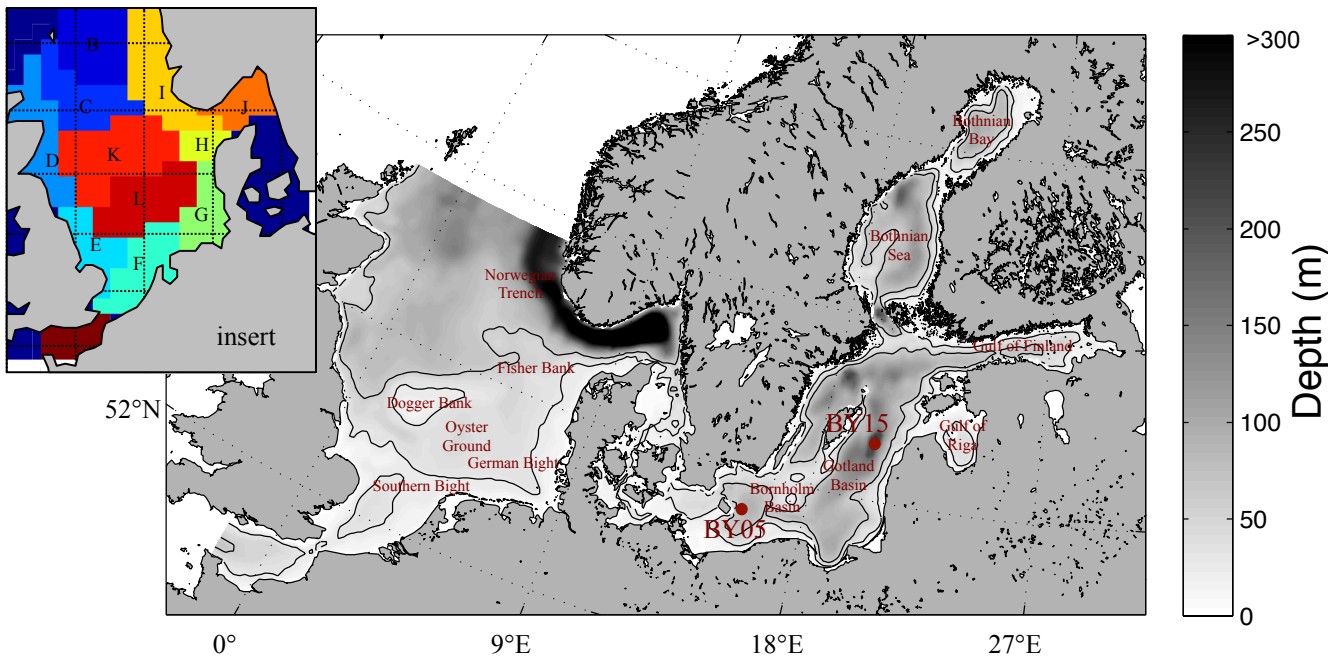

**Figure 1: Model area and bathymetry. Black lines indicate the 30 m and 60 m depth respectively. Insert: area subdivision in ICES-boxes for model comparison to ICES data (see Figure 10).**

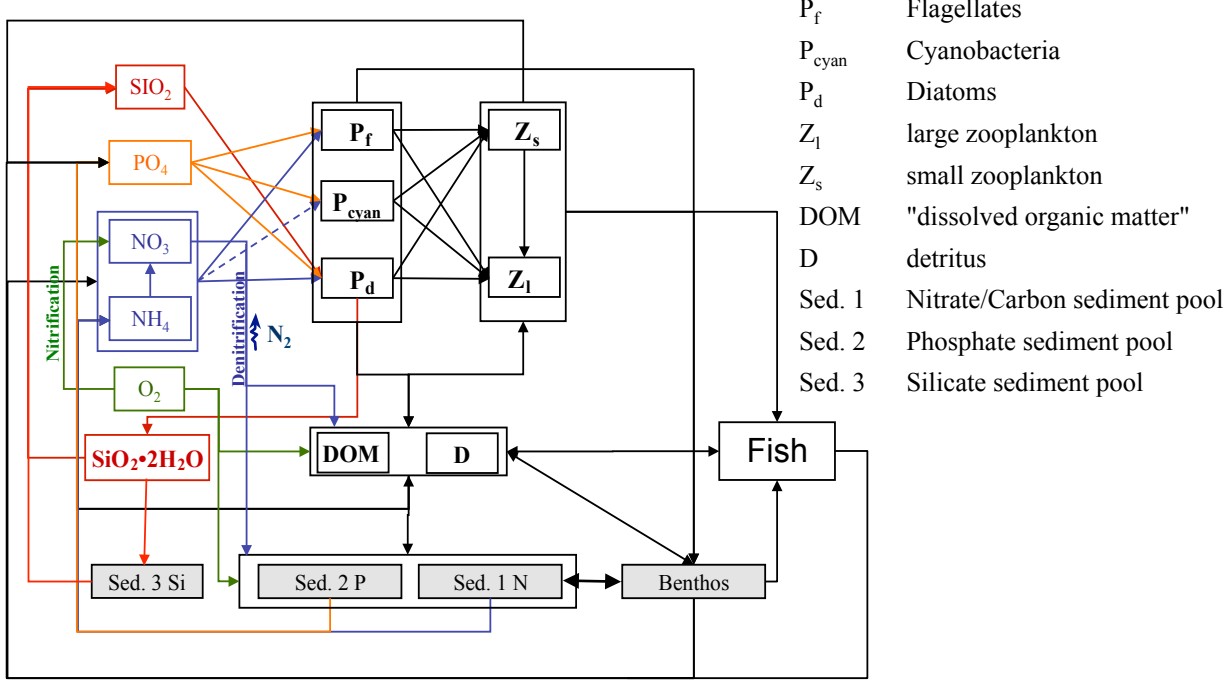

| | |
|---|---|
| $P_f$ | Flagellates |
| $P_{cyan}$ | Cyanobacteria |
| $P_d$ | Diatoms |
| $Z_l$ | large zooplankton |
| $Z_s$ | small zooplankton |
| DOM | "dissolved organic matter" |
| D | detritus |
| Sed. 1 | Nitrate/Carbon sediment pool |
| Sed. 2 | Phosphate sediment pool |
| Sed. 3 | Silicate sediment pool |

**Figure 2: Schematic diagram of biological-geochemical interactions in ECOSMO E2E.**

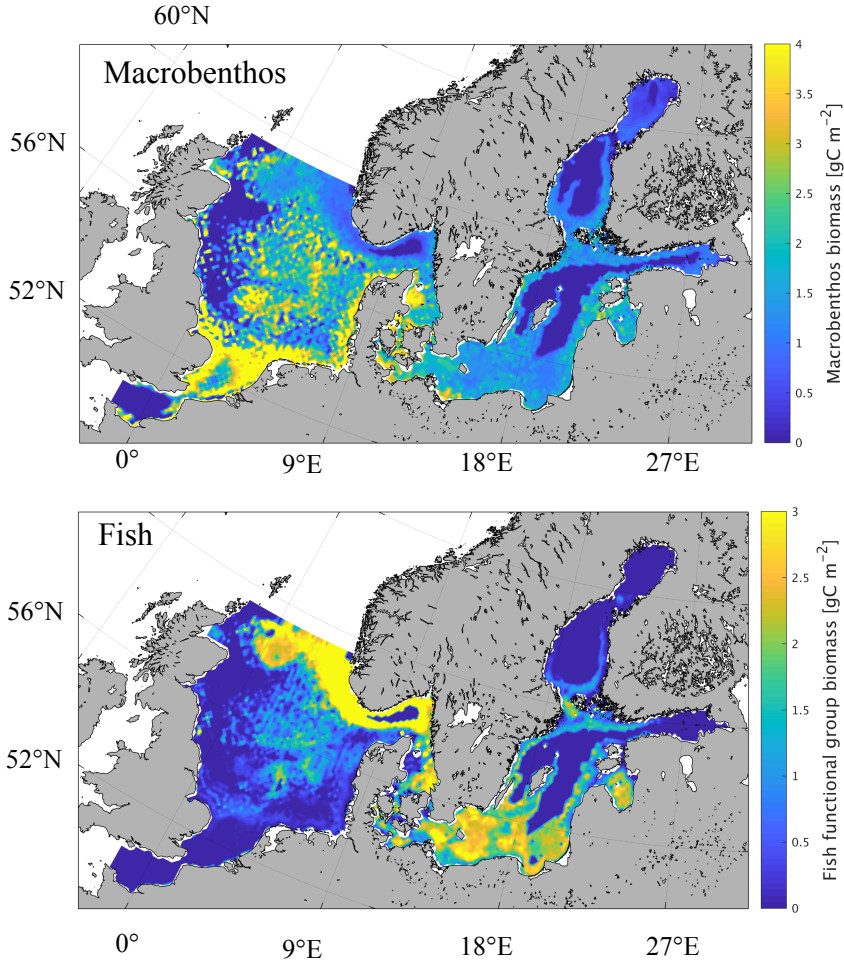

**Figure 3: Simulated spatial pattern of annual mean biomass of macrobenthos (upper panel) and fish (lower panel) (gC m⁻²).**

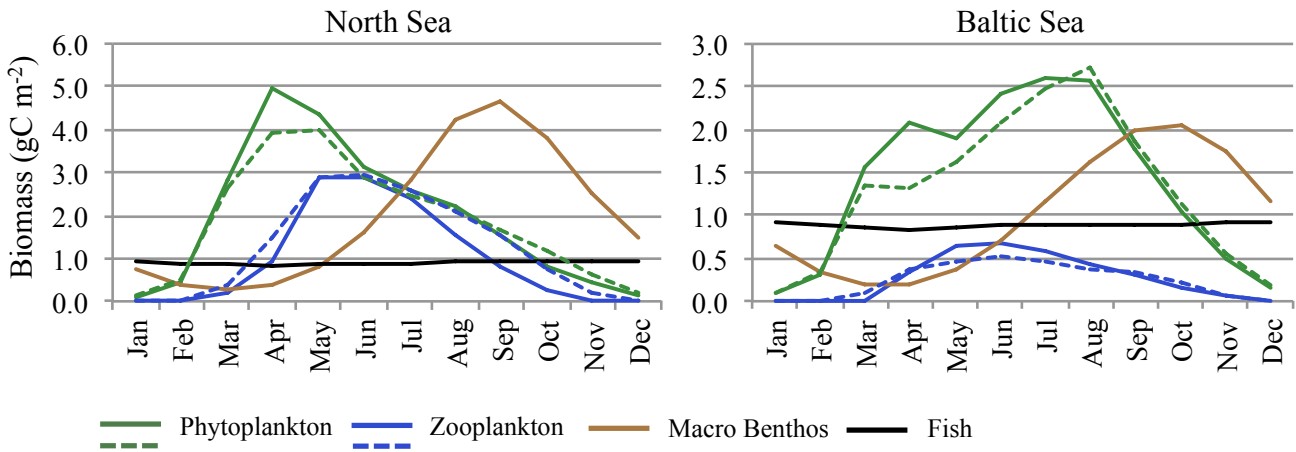

**Figure 4: Average seasonality of ecosystem components. Monthly means averaged for 1980-1989. solid lines: ECOSMO-E2E; dashed lines: ECOSMO II (phytoplankton & zooplankton).**

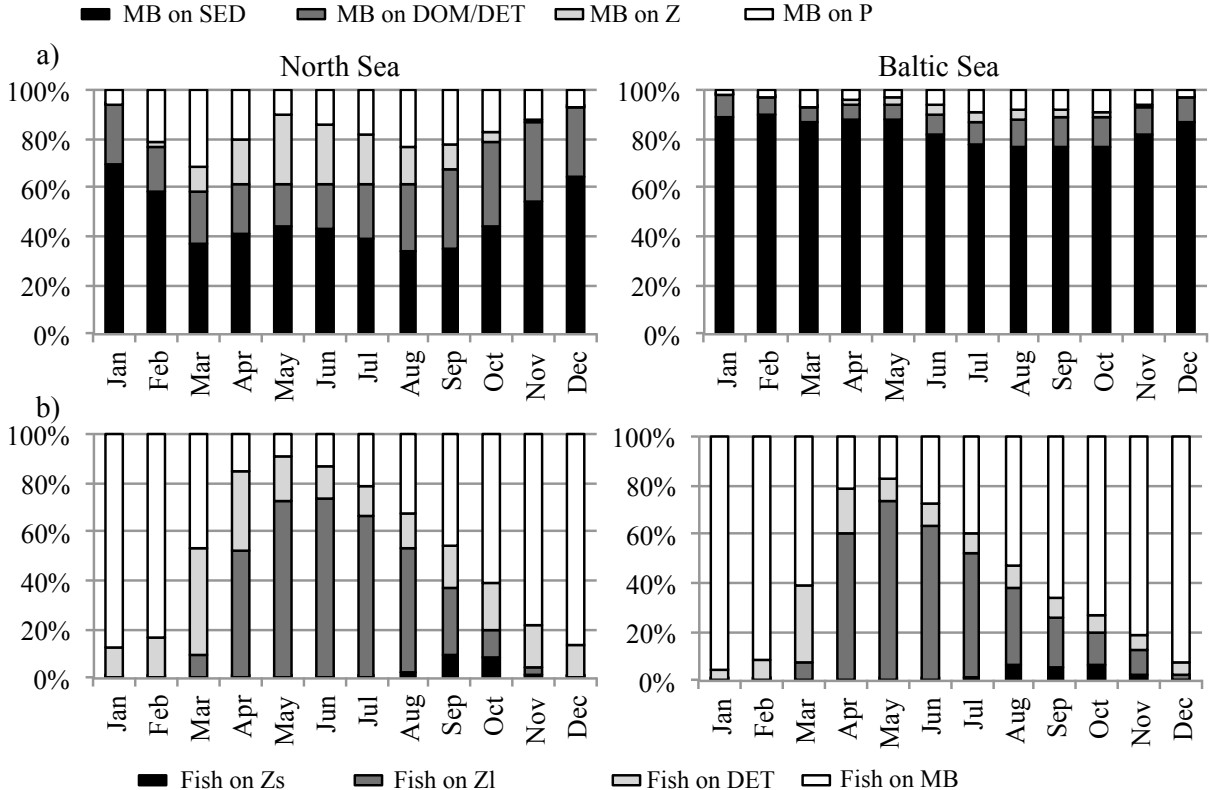

**Figure 5: Prey composition of a) macrobenthos (MB) and b) fish in the North Sea (left) and the Baltic Sea (right). MB feeds on SED (organic material in the sediment), DOM/DET (dead organic material in the water column: dissolved organic matter/detritus), Z (zooplankton), P (phytoplankton). Fish feeds on Zs ("small" herbivorous zooplankton), Zl ("large" omnivorous zooplankton), DET (detritus),.**

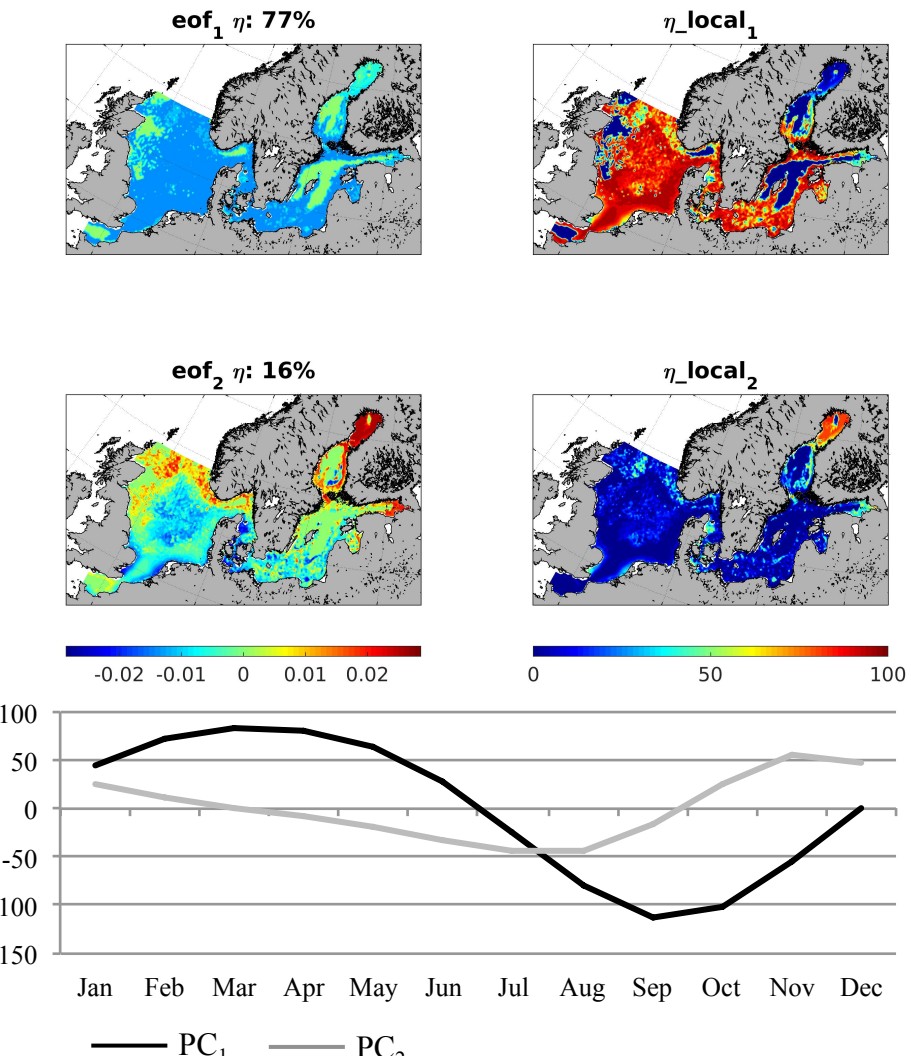

**Figure 6: Empirical Orthogonal Function (EOF) of macrobenthos biomass average seasonality (1980-1989). eof$_{1,2}$: spatial pattern of the first and second EOF mode; η: global explained variance; η_local: local explained variance for the first and second EOF; PC$_{1,2}$: temporal variability related to eof$_{1,2}$.**

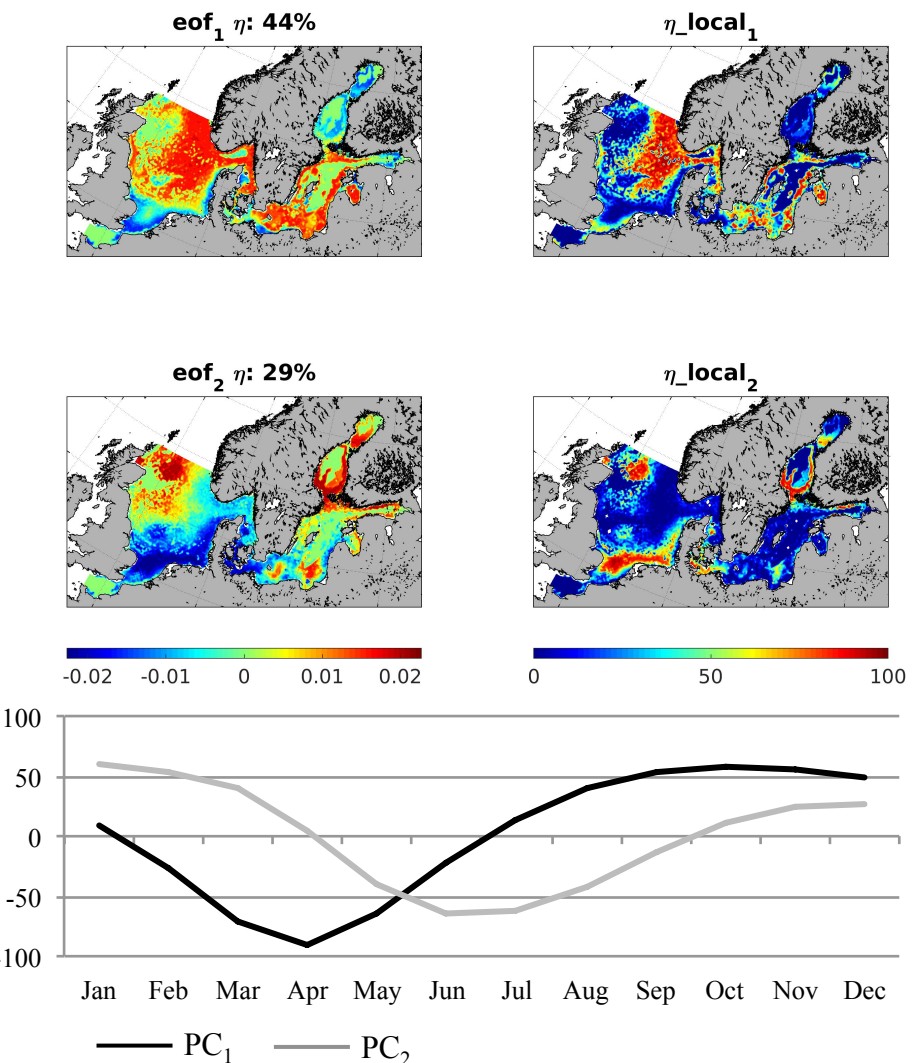

**Figure 7: Empirical Orthogonal Function (EOF) of fish biomass average seasonality (1980-1989). eof$_{1,2}$:spatial pattern of the first and second EOF mode; η: global explained variance; η_local: local explained variance for the first and second EOF; PC$_{1,2}$: temporal variability related to eof$_{1,2}$.**

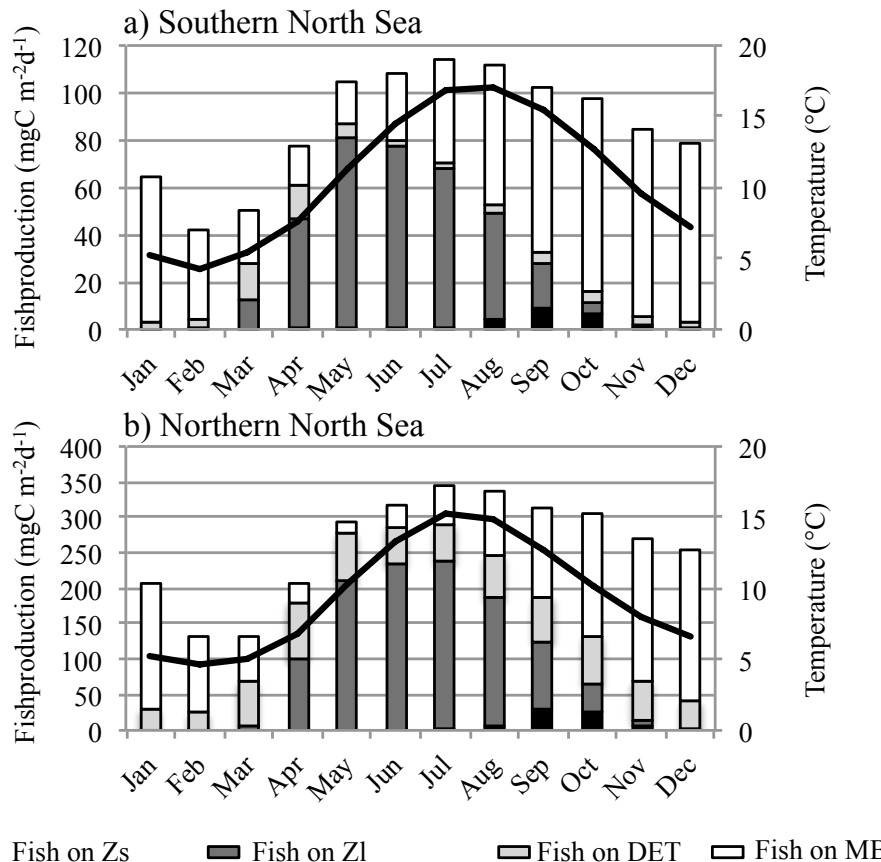

**Figure 8: Seasonal cycle on fish production (primary y-axis) in the shallow (depth<50m) southern North Sea (a) and in the deeper (depth> 50m) northern North Sea, divided into diet components (Zs: herbivorous zooplankton; Zl: omnivorous zooplakton; DET: detritus; MB: macrobenthos). And mean depth averaged temperature in the respective region (solid line; secondary y-axis).**

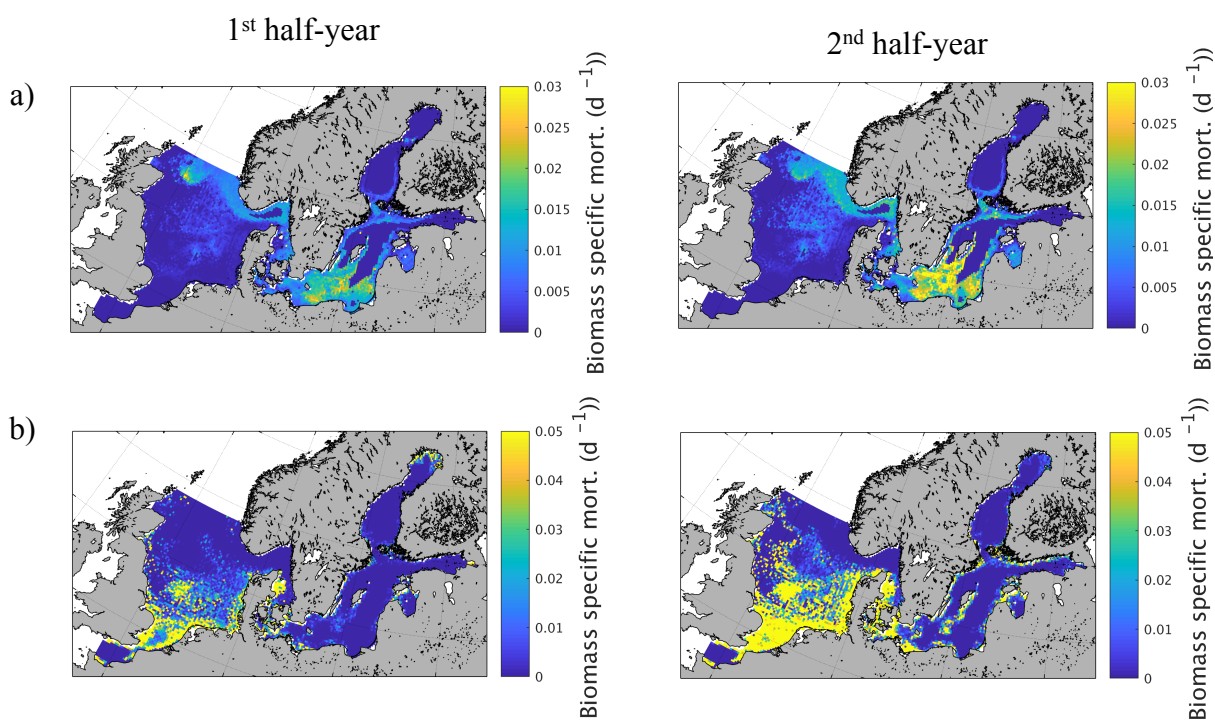

**Figure 9: Half-annually averaged (1980-1989) biomass specific mortality (d⁻¹) of zooplankton due to a) fish predation and b) macrobenthos predation.**

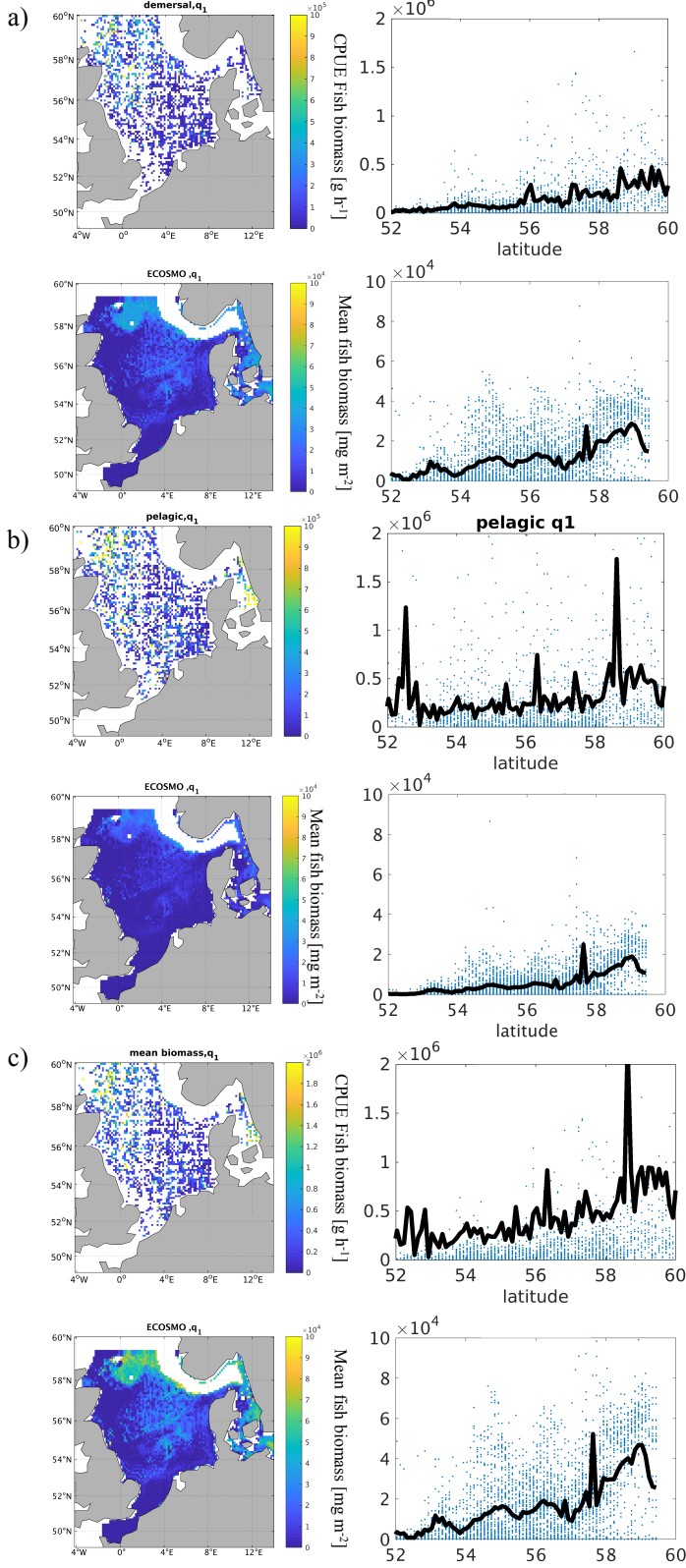

**Figure 10: a-c) Mean (1980-1989) total fish biomass from the ICES IBTS survey & fish biomass from ECOSMO E2E for the associated sampling time and area in the first quarter of the year (Jan-Mar) for demersal species (a), pelagic species (b), and combined biomass (c). Left panels: spatial distribution of fish biomass. Right panels: Biomass versus latitude and the mean of biomass at latitude (black line).**

Model Validation ICES Data

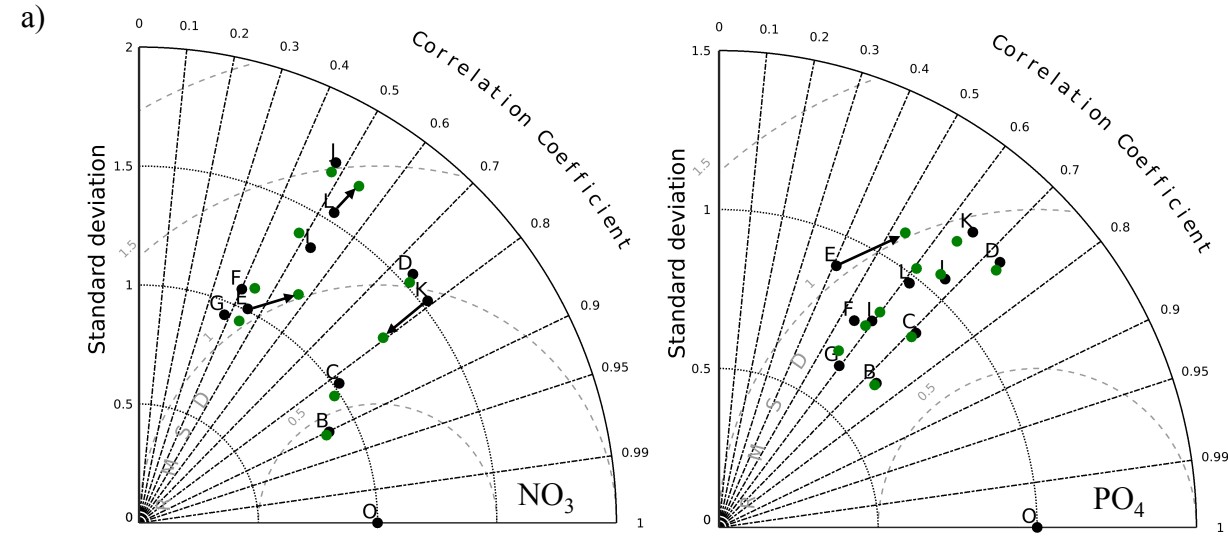

• ECOSMO  • ECOSMO-Fish

**Figure 9: Taylor diagram for surface (<10m) nutrients (model versus ICES data) in different areas of the North Sea (area separation in ICES boxes according to Figure 1) (nitrate (a); phophate (b)). Arrows indicate regions with relatively large changes in the validation measures.**

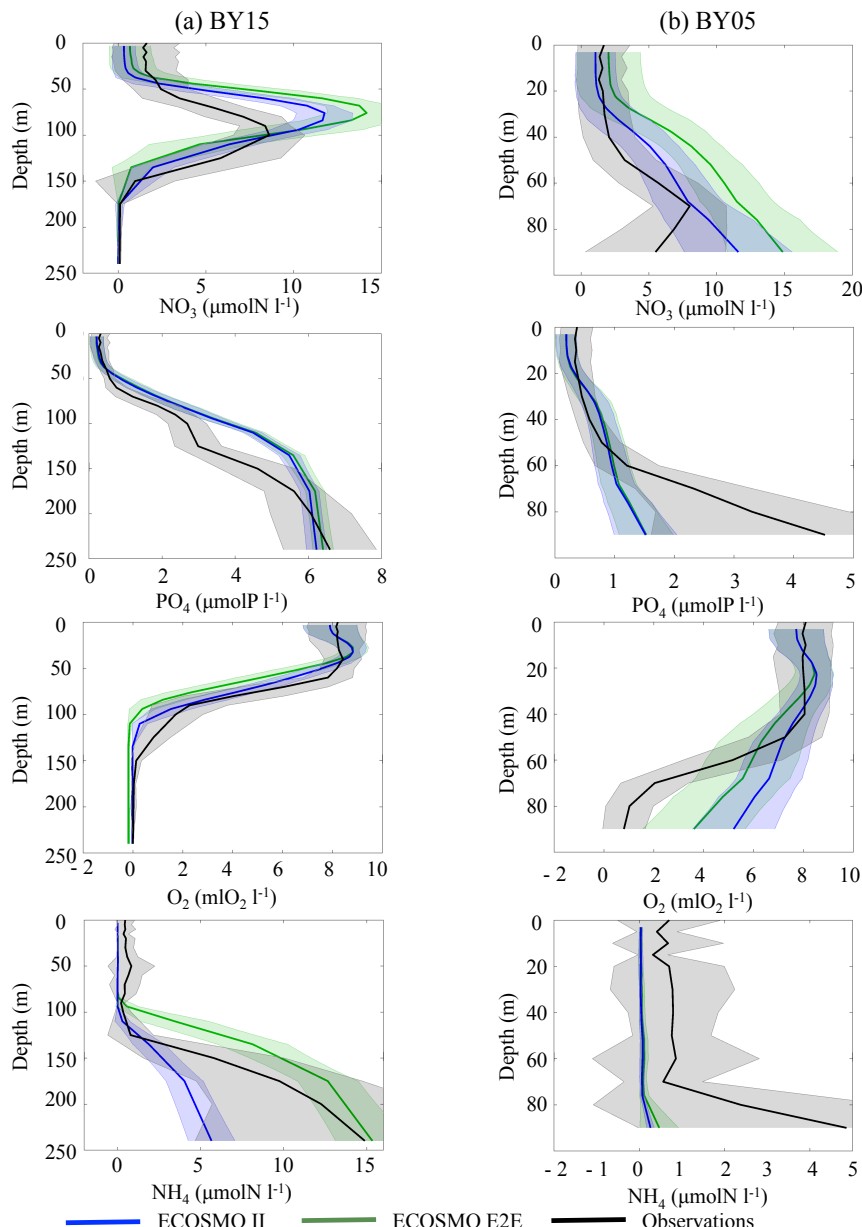

**Figure 10: Modeled (blue: ECOSMO II; green ECOSMO E2E) and observed (HELCOM data, black) vertical nutrient profiles. Data were averaged over the 10-year period 1980-1989 and mean (full line) and standard deviation (dashed line) are presented at two distinct locations in the Baltic Sea (BY15 (a); BY5 (b) see Figure 1).**

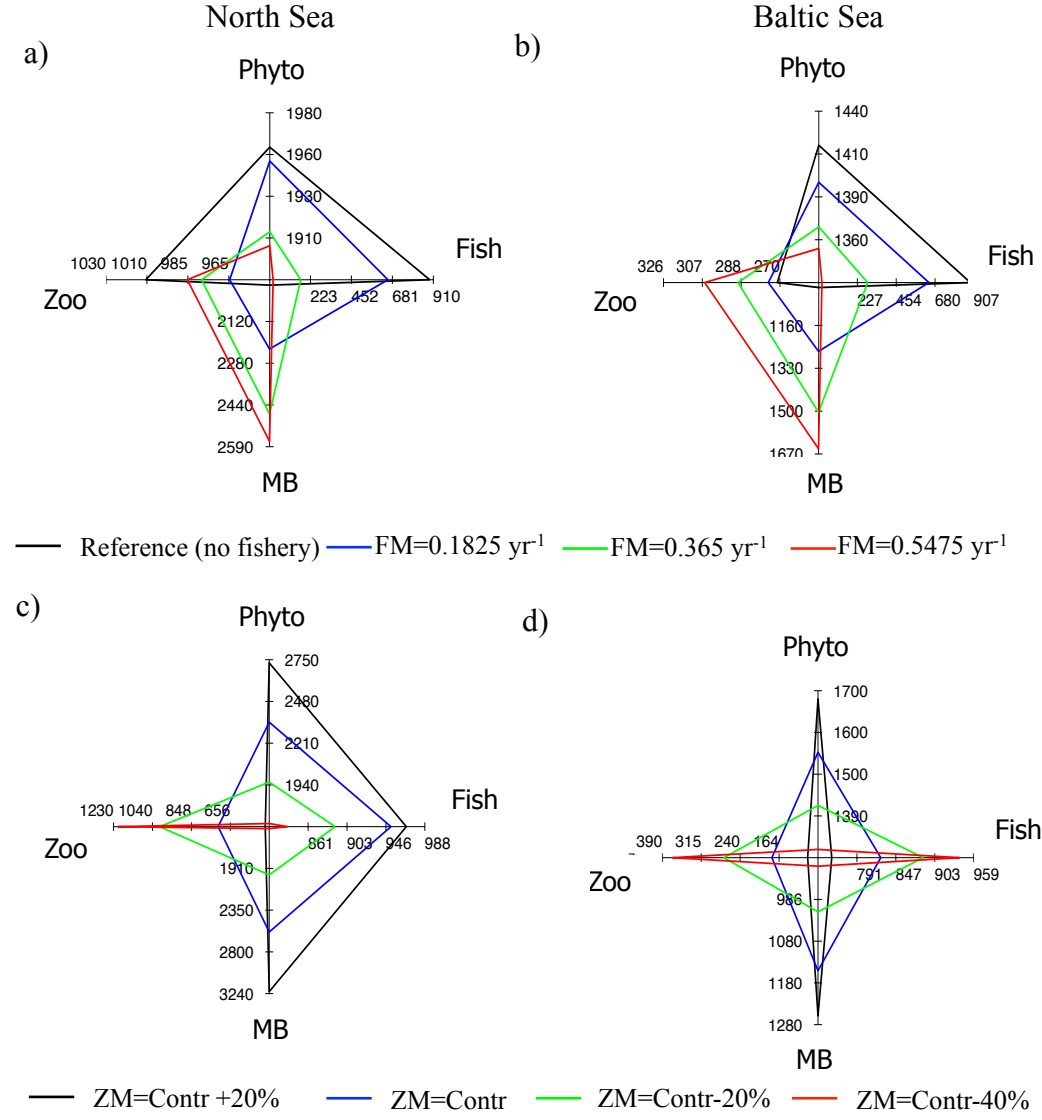

**Figure 11: Spider plots showing averaged changes in North Sea and Baltic Sea annual mean phytoplankton, zooplankton, MB, and fish biomass (mgC m$^2$) due to specific changes in the food web components. a,b: scenarios for fisheries mortality (FM); c,d: scenarios for changes in zooplankton natural mortality (ZM) .**

