# Peer review of "Towards End-2-End modelling in a consistent NPZD-F modelling framework (ECOSMOE2E\_vs1.0): Application to the North Sea and Baltic Sea"

_Geoscientific Model Development, 2018_

## Referee Comment (RC1) · Radtke (Referee) · 19 Dec 2018

On the example of the North and Baltic Sea, the authors present a very simple way to extend a lower trophic level model (marine biogeochemistry) to an end-to-end model. Two state variables have to be added, both of them two-dimensional (only horizontally resolved). These are the biomasses of fish and macrozoobenthos.

Since these two "upper trophic levels" influence especially the mortality of zooplankton, the result is a spatially heterogenous mortality which replaces the previously uniform closure term of the LTL model. A comparison to a study for the North Sea where fish predation mortality is estimated from observed distributions of planktivorous fish shows, as the authors state, that the correct spatial patterns emerge. In this way, the model outlines a rather simple way to include, to a first order, higher trophic level effects onto the lower trophic food web. Compared to the previous study, this approach has much lower data requirements, which is an advantage not pointed out well enough.

When it comes to overall model performance, the authors could show a slight benefit when adding the two state variables as they compare lower trophic level variables such as nutrient concentrations and oxygen. At least for the North Sea, the authors could also show that fish and macrozoobenthos spatial patterns were similar to observations.

From my point of view, the largest benefit of the model is the opportunity to gain more realistic zooplankton mortality values. I would like this point to be more highlighted. Especially I would like to see the spatial patterns of zooplankton mortality as they emerge from this model compared to those of Maar et al. (2014). Contrary to what the authors state in the caption of Fig. 9, fish production is not a suitable indicator for zooplankton predation mortality, but only the ratio between fish production and zooplankton biomass gives the predation mortality. The comparison should be done to Fig. 10C+D in Maar et al., not to Fig. 4, and it would be very useful if the comparison was shown in this article.

The discussion and conclusions sound reasonable to me, I would support them. The only exception is that I would not present total fish biomass as a model result, since it might be quite sensitive to the feeding efficiency which is poorly confined for the "average fish" I guess? I would rather interpret fish biomass as a tuning parameter which you have to fit to ensure that the ratio between predation mortality and background mortality is reasonable. In this way, quite simple measured quantities like total fish biomass and fishing mortality can be used to confine the zooplankton mortality, which was an arbitrary closure term before your extension.

**General remarks:**

1.) You say you can interpret the fish functional group as a "fish production potential". In principle it is clear what you mean: Where there is food, there will be fish who eat it. You are not interested in where they will migrate afterwards, but you keep the fish biomass locally as a kind of bookkeeping of what the fish consumed here. However, see a few issues with this interpretation:

- Zooplankton or macrobenthos will not be consumed if the fish just "potentially" grow. So, the loss terms for these functional groups contradict this interpretation.
- Consumption of food by fish is limited by local fish biomass in the model. A "fish production potential" in reality could be larger, since fish from remote locations could migrate towards a spot of high prey abundance, which might lead to more efficient food consumption.

I would like a clearer discussion of why neither of the two interpretations (fish biomass / fish production potential) is entirely correct.

2.) Your extension has a very low computational cost. Not only is it just two state variables that are added, but also these are 2-d only which saves their advection, and the advection of the state variables is the most time consuming step. So your extension is especially suitable for LTL models which are simple by purpose, e.g. because they are used in long-term climate simulations where computational load is critical. You could highlight that even more.

3.) You mix up British and American English, e.g. P1L20: "analysed", P1L24: "summarizes". Please be consistent.

4.) Also, please be consistent with "3d" vs. "3-d" / "end-to-end" vs. "End-to-End".

5.) Please capitalise "Figure 4" etc.

6.) Commas are often missing in sentences which do not start with the subject, please add them.

**Specific comments:**

P1L20: "the observed pattern" -> "the observed patterns"?

P1L26: "pattern agree" -> "patterns agree"?

P2L10: "The differentiation of trophic levels" -> hard to understand what exactly you mean, can you rephrase?

P2L11: Citation missing in reference list.

P2L26: Please explain "foodweb models" as opposed to end-2-end models

P2L31: "based on environmental condition" -> "based on environmental conditions"?

P2L31: "excluded" -> "excludes"?

P3L25: "relative low" -> "relatively low"?

P4L16-17: This is not a complete sentence.

P4L21-22: Please cite Neumann and Schernewski (2008) who invented this approach in the model world.

P5L2-3: "with additional restriction as" -> "with the additional restriction that"?

P5L8,10: Please use different symbols for the velocity vector and its vertical component.

P5L10: $A_v$ needs italics here. Actually, italics are missing a few times in this section when variables appear in the text. Please check.

P5L15: "RC" needs formatting.

P5L25: "the MB menu" -> "the MB diet"?

P6L2: I suggest the use of X' instead of X to make clear it has nothing to do with the X in line 1.

P6L29: To be precise, the vertical integral of equation 1 reduces to the equation given in the text. Equation 1 itself would still keep a vertical migration term $(C w_m(z))_z$, even if the vertical migration velocity $w_m(z)$ is only known implicitly.

P7L13: Does this consumption even occur in anoxic layers?

P7L14: There is no respiration of fish in the model? You rather treat fish respiration as an excretion and subsequent detritus mineralisation? Could you state this explicitly?

P7L20: In your formulation, TK is always equal to 1/273.15 because T/T cancels out. This is certainly not correct.

P7L28-29: Which food web did the study consider? I assume this value differs a lot between different seas/regions.

P7L31: "is considerable higher" -> "is considerably higher with"?

P7L31: Please state more explicitly that intraguild predation in zooplankton does not need to be represented in the model since it is not stage-resolving.

P8L21-27: Could you add a reference to the hydrodynamic model you used and the atmospheric forcing dataset?

P9L2: "at each of the location" -> "at each location" / "at each of the locations"?

P9L9: "on" -> "onto"

P10L6: "equals" -> "equals to"

P13L9: "concrete" -> "actual"?

P13L12: "relative small" -> "relatively small"?

P13L11-13: Could you give relative values (in % of average mass)?

P13L15: A minimum never falls, does it? A function of time falls until it reaches the minimum.

P13L30-31: Macrozoobenthos and fish can feed on their own excretions, since the energy contained in the carbon is not considered, correct? Also, there is no MB or fish respiration in the model which would convert the organic carbon to DIC. So, there can be an infinite loop in the model where MB feeds and excretes, feeds and excretes, with no need to add energy from primary production. In Section 4 where you discuss future model improvements, could you comment on how to prevent this? I see, besides the way of explicitly defining "nutritional value classes" in detritus, the possibility to limit detritus consumption to ensure that at least a specific percentage of the total diet is fresh (non-detrital) material.

P13L33: This is because the North Sea is a tidal sea, correct? I would consider it as helpful for the non-European readers to state this difference here.

P13L33: "Zooplankton and Phytoplankton is included" -> "Zooplankton and phytoplankton are included"?

P15L5: "zooplankton form" -> "zooplankton forms"?

P16L6-9: I would consider this as the main benefit of your model: You can obtain reasonable spatial patterns in zooplankton mortality without requiring data on planktivorous fish abundance. I would point this out already in the abstract and see it as the main point why your approach should be used.

P16L12-16: Couldn't you split your model fish into pelagic and demersal "feeding groups" based on their diet-dependent vertical distribution and then compare the spatial patterns?

P16L25: What is zero at which boundary?

P17L8: "fish and MB is resolved" -> "are resolved"?

P17L12: "for e.g." -> "e.g.", P18L31: the same

---

## Referee Comment (RC2) · Anonymous Referee #2 · 16 Jan 2019

GMD-2018-239

Overall, this is a well written and interesting study linking a classic NPZD model to a single fish compartment. As a fisheries modeller and ecosystem modeller familiar with OSMOSE and Atlantic frameworks, I found the approach useful, however as the authors acknowledge, simplistic in its treatment of fish and fisheries. There is no mention of the impact that fisheries have on the ecosystem until the results section. There should be some introductory material about this as they are the biggest impact on fish populations, as many fisheries reduce more than half of fish biomass as a goal. I would also like to see a more quantitative approach to calibrating the model to observed

biomass of fish and fisheries catches. This data is spatially available, and a difference plot or map showing how well predicted vs observed fish biomasses compare in a spatially explicit analysis would be useful. I'm also a bit concerned about the huge seasonality in biomass of the macrobenthos, much of which I presume is dominated by macroinvertebrates which don't vary in biomass as much as plankton communities do seasonally. It's not clear to me why the Baltic and North Seas were combined into one model, as they exhibit very different environmental and fish production regimes. I think the closure terms where a lot of fish migration could be happening would be more important to focus on than connecting the two domains. I would also like to see a more detailed treatment of fisheries mortality in the model, as this data is readily available and will be a huge driver of fish biomass given the very long exploitation history of the North Sea.

Intro - You don't describe how this component contributes to the model and how the macrobenthos communities vary in the North and Baltic Seas. Suggest using 'macrobenthos' instead of 'MB' throughout. Generally, think less acronyms could be used throughout as it makes it harder to follow.

Results and Discussion - I suggest separating out your results first and then comparing to other studies. The way the two sections are intertwined makes it difficult to follow.

Specific comments

page 2, lines 6-8 - the most common ecosystem models are Ecopath with Ecosim models which organize fish based on a combination of functional groups, species groups, and age-structured of species groups (see Tittensor et al. 2018 GMD).

page 3, lines 16-18 - How can questions about food webs be tested when there is only one fish functional group? There are different trophic levels of fish that are harvested which exert different controls on the macro food web. For example, forage fish have been shown to be important prey for many higher trophic levels and their exploitation has different effects than harvesting top predators (see Smith et al. 2010 Science).

page 4, line 1 - Awkward start to the sentence, suggest restructuring.

page 4, lines 16-17 - Awkward sentence, suggest combining with previous sentence in parentheses.

page 6, line 25 - What are the constraints for vertical fish movement based on oxygen and temperature limitations? This is one constraint of using only one fish group as there is a lot of variability among fish species in sensitivity to environmental conditions.

page 7, lines 17-18 - This sentence can be combined with the previous sentence.

page 8, lines 1-12 - What about fisheries mortality on the fish compartment? Was this not included in the model? For the North Sea, this would comprise a significant proportion of total mortality.

page 10, lines 11-16 - This is the first mention of fisheries, if they are a component of the model, then they should have been included much earlier in the manuscript. How was the loss rate calculated? Much greater detail about the fisheries data that was used and how this was applied need to be included.

page 10, lines 18-23 - How was this value of 20% less loss rate derived? Any empirical data that supports this value?

page 12, lines 24-26 - This is a reason why it is important to consider functional groups/species/age classes/size classes when including fish in ecosystem models.

page 12, line 28 - page 13, line 9 - How do the fish biomass estimates compare to reported fisheries landings from the regions? Is there enough biomass to support known landings?

page 13, line 11 - Is it reasonable for macrobenthos to vary that much annually? I would have expected there to be a much more constant standing stock similarly to fish. I'm thinking about the macro invertebrate community which doesn't vary that much through time. page 14, line 13 - You haven't introduced this analysis in the methods section to

describe what it is.

---

## Author Comment (AC1) · 15 Feb 2019

*On the example of the North and Baltic Sea, the authors present a very simple way to extend a lower trophic level model (marine biogeochemistry) to an end-to-end model. Two state variables have to be added, both of them two-dimensional (only horizontally resolved). These are the biomasses of fish and macrozoobenthos.*
*Since these two "upper trophic levels" influence especially the mortality of zooplankton, the result is a spatially heterogenous mortality which replaces the previously uniform closure term of the LTL model. A comparison to a study for the North Sea where fish predation mortality is estimated from observed distributions of planktivorous fish shows, as the authors state, that the correct spatial patterns emerge. In this way, the model outlines a rather simple way to include, to a first order, higher trophic level effects onto the lower trophic food web. Compared to the previous study, this approach has much lower data requirements, which is an advantage not pointed out well enough.*

*When it comes to overall model performance, the authors could show a slight benefit when adding the two state variables as they compare lower trophic level variables such as nutrient concentrations and oxygen. At least for the North Sea, the authors could also show that fish and macrozoobenthos spatial patterns were similar to observations.*

*From my point of view, the largest benefit of the model is the opportunity to gain more realistic zooplankton mortality values. I would like this point to be more highlighted. Especially I would like to see the spatial patterns of zooplankton mortality as they emerge from this model compared to those of Maar et al. (2014). Contrary to what the authors state in the caption of Fig. 9, fish production is not a suitable indicator for zooplankton predation mortality, but only the ratio between fish production and zooplankton biomass gives the predation mortality. The comparison should be done*
*to Fig. 10C+D in Maar et al., not to Fig. 4, and it would be very useful if the comparison was shown in this article.*

> Daewel et al: We agree that this comparison is necessary. We have compiled biomass specific mortality based on out model results as an average from 1980-1989 and separated the year into 1st and 2nd half as done in Maar et al. (2014) (see suppl. Figure 1a). The results are around the same magnitude as calculated by Maar et al. (2014) and the structure resembles the structure they had calculated for the first half of the year 2001. The results from Maar et al (2014) showed a clear difference between half-year 1 and half-year 2 with decreased biomass specific mortality in the second half-year in the central North Sea. This difference between the 1st and 2nd half-year is not evident in our model results. However we found a clear difference in magnitude when comparing winter and summer season (not shown). The reasons for the discrepancies between our model results and Maar et al (2014) are presumably related to interannual variations in fish consumption, which are not considered in a 10 year average, the fact that migration is not considered in the model and thus restrict the spatial variation, and that our functional group cannot resolve species and stage specific spatial and temporal variations like e.g. the increase in larval biomass in spring and changes in species composition. On the other hand, the approach from Maar et al. (2014) reveals uncertainties due to the fact that only parts of the North Sea fish assemblage is considered and that the fish biomass is prescribed and not dynamically coupled to zooplankton biomass. Considering the uncertainties in both approaches the comparison is actually quite good in both magnitude and spatial structure.
> Action: We suggest adding Suppl. Figure 1,(which also shows biomass specific zooplankton mortality due to macrobenthos), and a related discussion to the manuscript.

*The discussion and conclusions sound reasonable to me, I would support them. The only*

*exception is that I would not present total fish biomass as a model result, since it might be quite sensitive to the feeding efficiency which is poorly confined for the "average fish" I guess? I would rather interpret fish biomass as a tuning parameter which you have to fit to ensure that the ratio between predation mortality and background mortality is reasonable. In this way, quite simple measured quantities like total fish biomass and fishing mortality can be used to confine the zooplankton mortality, which was an arbitrary closure term before your extension.*

> Daewel et al.: We agree that showing total fish biomass as a model result is a bit critical in this context. On the other hand this is one of the few possibilities to confirm that the model is able to estimate reasonable results in terms of magnitude and distribution, otherwise it will certainly be difficult show that the approach actually improves the formulation of the predation mortality for zooplankton. So we would rather leave that part in the ms. We have added the scenarios simulations to the manuscript to identify the sensitivity of the results. However, we suggest clarifying the uncertainty of the variable more thoroughly in the ms, together with the response to general remark 1 below.

General remarks:
*1.) You say you can interpret the fish functional group as a "fish production potential". In principle it is clear what you mean: Where there is food, there will be fish who eat it. You are not interested in where they will migrate afterwards, but you keep the fish biomass locally as a kind of bookkeeping of what the fish consumed here. However, see a few issues with this interpretation:*
*· Zooplankton or macrobenthos will not be consumed if the fish just "potentially" grow. So, the loss terms for these functional groups contradict this interpretation.*
*· Consumption of food by fish is limited by local fish biomass in the model. A "fish production potential" in reality could be larger, since fish from remote locations could migrate towards a spot of high prey abundance, which might lead to more efficient food consumption. I would like a clearer discussion of why neither of the two interpretations (fish biomass / fish production potential) is entirely correct.*

> Daewel et al: Yes we agree with the reviewer, this needs to be clarified in the ms. We will add the following explanation to the method section: "In the following, we will thus refer to "fish" as a functional group that comprises the fish biomass that emerges based on the lower trophic production at each horizontal grid cell. For clarification it needs to be noted that, even when called "fish production potential", the fish biomass is a state variable in the model that interacts dynamically with the lower trophic level components and that will be used in the following to confirm the models ability to simulate spatial and temporal pattern of carbon transfer to higher trophic levels. On the other hand by constraining the horizontal migration capabilities of the fish group to one grid cell we will likely underestimate the local fish production potential by confining it to the locally available fish biomass."

*2.) Your extension has a very low computational cost. Not only is it just two state variables that are added, but also these are 2-d only which saves their advection, and the advection of the state variables is the most time consuming step. So your extension is especially suitable for LTL models which are simple by purpose, e.g. because they are used in long-term climate simulations where computational load is critical. You could highlight that even more.*

> Daewel et al.: Yes thank you. We will add this more clearly to the conclusion section.

*3.) You mix up British and American English, e.g. P1L20: "analysed", P1L24: "summarizes". Please be consistent.* –will be changed

*4.) Also, please be consistent with "3d" vs. "3-d" / "end-to-end" vs. "End-to-End".* –will be changed

5.) *Please capitalise "Figure 4" etc.* – will be changed

6.) *Commas are often missing in sentences which do not start with the subject, please add them.* – we will correct where commas are missing

Specific comments:

*P1L20: "the observed pattern" -> "the observed patterns"?* -ok

*P1L26: "pattern agree" -> "patterns agree"?* -ok

*P2L10: "The differentiation of trophic levels" -> hard to understand what exactly you mean, can you rephrase?* – Right! Changed to : "the separation of trophic levels ..."

*P2L11: Citation missing in reference list.* - added

*P2L26: Please explain "foodweb models" as opposed to end-2-end models*

> Daewel et al: Here, a food web model is a model that resolves a complex food web on the basis of species or very specific functional groups. In principle a food web model does not necessarily need to be End-2-End. However the model Atlantis is both a food web model and End-2-End. We will make that more clear in the ms.

*P2L31: "based on environmental condition" -> "based on environmental conditions"?* –will be changed

*P2L31: "excluded" -> "excludes"?* - will be changed

*P3L25: "relative low" -> "relatively low"?* - will be changed

*P4L16-17: This is not a complete sentence.* –combined with previous sentence

*P4L21-22: Please cite Neumann and Schernewski (2008) who invented this approach in the model world.* – will be added to the ms

*P5L2-3: "with additional restriction as" -> "with the additional restriction that"?-* will be changed

*P5L8,10: Please use different symbols for the velocity vector and its vertical component.* – will be changed

*P5L10: A_v needs italics here. Actually, italics are missing a few times in this section when variables appear in the text. Please check.* – will be changed

*P5L15: "RC" needs formatting.* -will be changed

P5L25: "the MB menu" -> "the MB diet"? – will be changed

*P6L2: I suggest the use of X' instead of X to make clear it has nothing to do with the X in line 1.* – The X in line 2 is the same X as in line 1. However, X was not clarified in the text. We suggest modifying the text as follows:

"Grazing rates GMB on prey type X ($X \epsilon [Z_1; Z_2; P_1; P_2; DET; DOM; SED1]$) are estimated using ..."

*P6L29: To be precise, the vertical integral of equation 1 reduces to the equation given in the text. Equation 1 itself would still keep a vertical migration term (C w_m(z))_z, even if the vertical migration velocity w_m(z) is only known implicitly.*

> Daewel et al: Yes right that is not explained correctly. We suggest changing the sentence to: "Following those three principles implies that equation 1 is simplified to $\frac{\partial C_{Fi}}{\partial t} + w_m(z) \frac{\partial C_{Fi}}{\partial z} = R_{Fi}$, where $w_m(z)$ is the vertical migration speed, which is given implicitly by the dynamical vertical distribution of the fish biomass in dependence of the vertical prey distribution. ... „

*P7L13: Does this consumption even occur in anoxic layers?* – No, fish consumption is confined to oxic conditions. We will add the information to the ms.

*P7L14: There is no respiration of fish in the model? You rather treat fish respiration as an excretion and subsequent detritus mineralisation? Could you state this explicitly?*

> Daewel et al: The term excretion loss is related to respiration of MB and fish does not include fecal excretion. It is considered directly as a source of dissolved inorganic nutrient ($PO_4/NH_4$ in principle also DIC if that were a state variable of

the model)(see table 4 in the ms). See also explanation to P13L30-31 below. We will clarify that in the ms.

*P7L20: In your formulation, TK is always equal to 1/273.15 because T/T cancels out. This is certainly not correct.*

Daewel et al: You are right. Thanks for pointing this out. The equation given here is not correct. Will be changed to: $\varepsilon_{Fi} = \mu_{Fi}e^{\left(\frac{\theta_{Fi}}{k}*TK\right)}$; $TK = \frac{T-T_0}{T*T_0}$ with T is given in °K and $T_0$=273.15 °K.

*P7L28-29: Which food web did the study consider? I assume this value differs a lot between different seas/regions.* – The food web of the North Sea is considered in the study. - information will be added to the ms

*P7L31: "is considerable higher" -> "is considerably higher with"?* –will be changed

*P7L31: Please state more explicitly that intraguild predation in zooplankton does not need to be represented in the model since it is not stage-resolving.* – will be added to the ms.

*P8L21-27: Could you add a reference to the hydrodynamic model you used and the atmospheric forcing dataset?* – We will add more details to the setup paragraph.

*P9L2: "at each of the location" -> "at each location" / "at each of the locations"?* - will be changed

*P9L9: "on" -> "onto"* - will be changed

*P10L6: "equals" -> "equals to"* -changed

*P13L9: "concrete" -> "actual"?* -changed

*P13L12: "relative small" -> "relatively small"?* –changed

*P13L11-13: Could you give relative values (in % of average mass)?*

Daewel et al: Yes we will give these values. Following a suggestion by reviewer 2 we will also add a discussion on the MB seasonality to a revised version of the ms.

*P13L15: A minimum never falls, does it? A function of time falls until it reaches the minimum.* – Yes, right. Will be changed to "reaches"

*P13L30-31: Macrozoobenthos and fish can feed on their own excretions, since the energy contained in the carbon is not considered, correct? Also, there is no MB or fish respiration in the model, which would convert the organic carbon to DIC. So, there can be an infinite loop in the model where MB feeds and excretes, feeds and excretes, with no need to add energy from primary production. In Section 4 where you discuss future model improvements, could you comment on how to prevent this? I see, besides the way of explicitly defining "nutritional value classes" in detritus, the possibility to limit detritus consumption to ensure that at least a specific percentage of the total diet is fresh (non-detrital) material.*

Daewel et al: We understand your concern, but we cannot agree on this conclusion, and fear that the model description might be a bit unclear in that respect. On the one hand only parts of what is consumed can be assimilated while the rest can be considered as fecal excretion (this is probably what the reviewer is referring to). On the other hand the additional loss term for both fish and MB consists of a mortality and an excretion term. The mortality loss and fecal excretion contributes to the detritus pool, while the excretion loss is related to respiration of MB and fish and is considered directly as a source of dissolved inorganic nutrient ($PO_4$/$NH_4$ in principle also DIC if that were a state variable of the model)(see table 4 in the ms). Thus the feeding cycle features two specific loss term; i) the excretion term and ii) the remineralization of the dead organic matter, which would prevent the occurrence of such a self-containing feeding chain.

Action: We will clarify these details in the model description of the ms. We also found an error in the last equation of table 4, which will be corrected

*P13L33: This is because the North Sea is a tidal sea, correct? I would consider it as helpful for the non-European readers to state this difference here.*

> Daewel et al: Right, although the differences have been explained in the introduction, it is probably good to make that clear at this point.
> Action: we suggest changing the sentence to:" … presumably due to the fact that a higher percentage of detritus is re-suspended in the tidal influenced, highly turbulent areas of the North Sea."

*P13L33: "Zooplankton and Phytoplankton is included" -> "Zooplankton and phytoplankton are included"?* – will be changed

*P15L5: "zooplankton form" -> "zooplankton forms"?* – will be changed

*P16L6-9: I would consider this as the main benefit of your model: You can obtain reasonable spatial patterns in zooplankton mortality without requiring data on planktivorous fish abundance. I would point this out already in the abstract and see it as the main point why your approach should be used.*

> Daewel et al: Thanks for pointing that out. We will follow the suggestion and add this more prominent to the abstract of the revised ms.

*P16L12-16: Couldn't you split your model fish into pelagic and demersal "feeding groups" based on their diet-dependent vertical distribution and then compare the spatial patterns?*

> Daewel et al: We have tried to do so by differentiating between biomass in the water column and in the bottom layer (Suppl. Figure 2). However, clearly this is not directly comparable to the observation, where demersal and pelagic biomass is separated by species not actually by vertical appearance. Additionally the fish in our model has the chance to feed on all food sources at each time step. Nonetheless we agree that this comparison can be helpful and thus will include it in a revised version of the manuscript together with a discussion. One thing that becomes quite apparent is, as already suggested in the discussion, that the pelagic fish biomass is underrepresented by the model, since, in contrast to the observations, the bottom biomass exceeds that of the water column biomass. On the other hand, similar to the observation, both stocks contribute to the latitudinal gradient in biomass.
> Note that there was a unit error in the original version of the Figure 10d that was corrected in the new figure (Suppl. Figure 2).

*P16L25: What is zero at which boundary?* – Here we mean that no fish enters the area through the open boundaries to the North Atlantic by e.g. migration. Action: will be clarified in the ms

*P17L8: "fish and MB is resolved" -> "are resolved"?* - will be changed

*P17L12: "for e.g." -> "e.g.", P18L31: the same* - will be changed

[Figure]

Suppl. Figure 1 Half-annually averaged biomass specific mortality $(d^{-1})$ of zooplankton due to fish predation (upper panels) and macrobenthos predation (lower panels).

[Figure]

Suppl. Figure 2 Mean (1980-1989) total fish biomass as estimated from ECOSMO E2E . For biomass in the bottom layer a), in the pelagic water layers b), and the combined biomass. Left panels: spatial distribution of fish biomass. Right panels: Biomass versus latitude and the mean of biomass at latitude (black line).

---

## Author Response (AR1)

*Reply reviewer 1*

*On the example of the North and Baltic Sea, the authors present a very simple way to extend a lower trophic level model (marine biogeochemistry) to an end-to-end model. Two state variables have to be added, both of them two-dimensional (only horizontally resolved). These are the biomasses of fish and macrozoobenthos.*
*Since these two "upper trophic levels" influence especially the mortality of zooplankton, the result is a spatially heterogenous mortality which replaces the previously uniform closure term of the LTL model. A comparison to a study for the North Sea where fish predation mortality is estimated from observed distributions of planktivorous fish shows, as the authors state, that the correct spatial patterns emerge. In this way, the model outlines a rather simple way to include, to a first order, higher trophic level effects onto the lower trophic food web. Compared to the previous study, this approach has much lower data requirements, which is an advantage not pointed out well enough.*

*When it comes to overall model performance, the authors could show a slight benefit when adding the two state variables as they compare lower trophic level variables such as nutrient concentrations and oxygen. At least for the North Sea, the authors could also show that fish and macrozoobenthos spatial patterns were similar to observations.*

*From my point of view, the largest benefit of the model is the opportunity to gain more realistic zooplankton mortality values. I would like this point to be more highlighted. Especially I would like to see the spatial patterns of zooplankton mortality as they emerge from this model compared to those of Maar et al. (2014). Contrary to what the authors state in the caption of Fig. 9, fish production is not a suitable indicator for zooplankton predation mortality, but only the ratio between fish production and zooplankton biomass gives the predation mortality. The comparison should be done*
*to Fig. 10C+D in Maar et al., not to Fig. 4, and it would be very useful if the comparison was shown in this article.*

Daewel et al: We agree that this comparison is necessary. We have compiled biomass specific mortality based on out model results as an average from 1980-1989 and separated the year into 1st and 2nd half as done in Maar et al. (2014) (see suppl. Figure 1a). The results are around the same magnitude as calculated by Maar et al. (2014) and the structure resembles the structure they had calculated for the first half of the year 2001. The results from Maar et al (2014) showed a clear difference between half-year 1 and half-year 2 with decreased biomass specific mortality in the second half-year in the central North Sea. This difference between the 1st and 2nd half-year is not evident in our model results. However we found a clear difference in magnitude when comparing winter and summer season (not shown). The reasons for the discrepancies between our model results and Maar et al (2014) are presumably related to interannual variations in fish consumption, which are not considered in a 10 year average, the fact that migration is not considered in the model and thus restrict the spatial variation, and that our functional group cannot resolve species and stage specific spatial and temporal variations like e.g. the increase in larval biomass in spring and changes in species composition. On the other hand, the approach from Maar et al. (2014) reveals uncertainties due to the fact that only parts of the North Sea fish assemblage is considered and that the fish biomass is prescribed and not dynamically coupled to zooplankton biomass. Considering the uncertainties in both approaches the comparison is actually quite good in both magnitude and spatial structure.
Action: We replaced the old Figure 9 with Suppl. Figure 1 (now the new Figure 9),(which also shows biomass specific zooplankton mortality due to macrobenthos), and substantially modified the text and discussion related to

Figure 9.

*The discussion and conclusions sound reasonable to me, I would support them. The only exception is that I would not present total fish biomass as a model result, since it might be quite sensitive to the feeding efficiency which is poorly confined for the "average fish" I guess? I would rather interpret fish biomass as a tuning parameter which you have to fit to ensure that the ratio between predation mortality and background mortality is reasonable. In this way, quite simple measured quantities like total fish biomass and fishing mortality can be used to confine the zooplankton mortality, which was an arbitrary closure term before your extension.*

> Daewel et al.: We agree that showing total fish biomass as a model result is a bit critical in this context. On the other hand, this is one of the few possibilities to confirm that the model is able to estimate reasonable results in terms of magnitude and distribution, otherwise it will certainly be difficult show that the approach actually improves the formulation of the predation mortality for zooplankton. So we would rather leave that part in the ms. We have added the scenarios simulations to the manuscript to identify the sensitivity of the results. However, we suggest clarifying the uncertainty of the variable more thoroughly in the ms, together with the response to general remark 1 below.

General remarks:

*1.) You say you can interpret the fish functional group as a "fish production potential". In principle it is clear what you mean: Where there is food, there will be fish who eat it. You are not interested in where they will migrate afterwards, but you keep the fish biomass locally as a kind of bookkeeping of what the fish consumed here. However, see a few issues with this interpretation:*
*· Zooplankton or macrobenthos will not be consumed if the fish just "potentially" grow. So, the loss terms for these functional groups contradict this interpretation.*
*· Consumption of food by fish is limited by local fish biomass in the model. A "fish production potential" in reality could be larger, since fish from remote locations could migrate towards a spot of high prey abundance, which might lead to more efficient food consumption. I would like a clearer discussion of why neither of the two interpretations (fish biomass / fish production potential) is entirely correct.*

> Daewel et al: Yes we agree with the reviewer, this needs to be clarified in the ms. We will add the following explanation to the method section: "In the following, we will thus refer to "fish" as a functional group that comprises the fish biomass that emerges based on the lower trophic production at each horizontal grid cell. For clarification it needs to be noted that, even when called "fish production potential", the fish biomass is a state variable in the model that interacts dynamically with the lower trophic level components and that will be used in the following to confirm the models ability to simulate spatial and temporal pattern of carbon transfer to higher trophic levels. On the other hand by constraining the horizontal migration capabilities of the fish group to one grid cell we will likely underestimate the local fish production potential by confining it to the locally available fish biomass."

*2.) Your extension has a very low computational cost. Not only is it just two state variables that are added, but also these are 2-d only which saves their advection, and the advection of the state variables is the most time consuming step. So your extension is especially suitable for LTL models which are simple by purpose, e.g. because they are used in long-term climate simulations where computational load is critical. You could highlight that even more.*

> Daewel et al.: Yes thank you. We will add this more clearly to the conclusion section.

*3.) You mix up British and American English, e.g. P1L20: "analysed", P1L24: "summarizes". Please be consistent.* –changed

*4.) Also, please be consistent with "3d" vs. "3-d" / "end-to-end" vs. "End-to-End".* –will be changed

5.) *Please capitalise "Figure 4" etc.* –changed

6.) *Commas are often missing in sentences which do not start with the subject, please add them.* –corrected where commas are missing

Specific comments:

*P1L20: "the observed pattern" -> "the observed patterns"?* -ok

*P1L26: "pattern agree" -> "patterns agree"?* -ok

*P2L10: "The differentiation of trophic levels" -> hard to understand what exactly you mean, can you rephrase? –* Right! Changed to : "the separation of trophic levels ..."

*P2L11: Citation missing in reference list.* - added

*P2L26: Please explain "foodweb models" as opposed to end-2-end models*

> Daewel et al: Here, a food web model is a model that resolves a complex food web on the basis of species or very specific functional groups. In principle a food web model does not necessarily need to be End-2-End. However the model Atlantis is both a food web model and End-2-End. We made that more clear in the ms.

*P2L31: "based on environmental condition" -> "based on environmental conditions"? –* changed

*P2L31: "excluded" -> "excludes"?* - changed

*P3L25: "relative low" -> "relatively low"?* - changed

*P4L16-17: This is not a complete sentence.* –combined with previous sentence

*P4L21-22: Please cite Neumann and Schernewski (2008) who invented this approach in the model world.* –added to the ms

*P5L2-3: "with additional restriction as" -> "with the additional restriction that"?-* changed

*P5L8,10: Please use different symbols for the velocity vector and its vertical component. –* changed

*P5L10: A_v needs italics here. Actually, italics are missing a few times in this section when variables appear in the text. Please check.* –changed

*P5L15: "RC" needs formatting.* - changed

P5L25: "the MB menu" -> "the MB diet"? –changed

*P6L2: I suggest the use of X' instead of X to make clear it has nothing to do with the X in line 1. –* The X in line 2 is the same X as in line 1. However, X was not clarified in the text. We modified the text as follows:

"Grazing rates GMB on prey type X ($X \in [Z_1; Z_2; P_1; P_2; DET; DOM; SED1]$) are estimated using ..."

*P6L29: To be precise, the vertical integral of equation 1 reduces to the equation given in the text. Equation 1 itself would still keep a vertical migration term (C w_m(z))_z, even if the vertical migration velocity w_m(z) is only known implicitly.*

> Daewel et al: Yes right that is not explained correctly. We changed the sentence to: "Following those three principles implies that equation 1 is simplified to $\frac{\partial C_{Fi}}{\partial t} + w_m(z) \frac{\partial C_{Fi}}{\partial z} = R_{Fi}$, where $w_m(z)$ is the vertical migration speed, which is given implicitly by the dynamical vertical distribution of the fish biomass in dependence of the vertical prey distribution. ... „

P7L13: *Does this consumption even occur in anoxic layers*? – No, fish consumption is confined to oxic conditions. We added the following sentence to the ms: "Note that, since fish do not tolerate anoxic conditions, only grid cells featuring positive oxygen concentrations were considered for the estimate of fish consumption."

*P7L14: There is no respiration of fish in the model? You rather treat fish respiration as an excretion and subsequent detritus mineralisation? Could you state this explicitly?*

> Daewel et al: The term excretion loss is related to respiration of MB and fish and does not include fecal excretion. It is considered directly as a source of dissolved inorganic nutrient ($PO_4/NH_4$ in principle also DIC if that were a state variable of the model)(see table 4 in the ms). See also explanation to P13L30-31 below. Action: the role of the excretion term has now been clarified in the model description. We also modified table 4 since the reaction term for oxygen was not given in the earlier version of the ms.

*P7L20: In your formulation, TK is always equal to 1/273.15 because T/T cancels out. This is certainly not correct.*

> Daewel et al: You are right. Thanks for pointing this out. The equation given here is not correct. Will be changed to: $\varepsilon_{Fi} = \mu_{Fi}e^{\left(\frac{\theta_{Fi}}{k}*TK\right)}$; $TK = \frac{T-T_0}{T*T_0}$ with T is given in °K and $T_0$=273.15 °K.
> Action: Equ. 11 has been changed.

*P7L28-29: Which food web did the study consider? I assume this value differs a lot between different seas/regions.* – The food web of the North Sea is considered in the study. - information has been added to the ms
*P7L31: "is considerable higher" -> "is considerably higher with"?* –changed
*P7L31: Please state more explicitly that intraguild predation in zooplankton does not need to be represented in the model since it is not stage-resolving.* –added to the ms:"Since the zooplankton model is not stage resolving intraguild predation is not explicitly prescribed as a mortality term, but is implicitly included in the background mortality."

*P8L21-27: Could you add a reference to the hydrodynamic model you used and the atmospheric forcing dataset?* – More detail was added to the setup paragraph.
*P9L2: "at each of the location" -> "at each location" / "at each of the locations"?* - changed
*P9L9: "on" -> "onto"* - changed
*P10L6: "equals" -> "equals to"* -changed
*P13L9: "concrete" -> "actual"?* -changed
*P13L12: "relative small" -> "relatively small"?* –changed
*P13L11-13: Could you give relative values (in % of average mass)?*

> Daewel et al: Yes we will give these values. "Values for both MB and fish biomass vary in the course of the year (Fig. 4). While the modelled seasonal amplitude for fish biomass is relatively small (North Sea: 94.7 mgC m-3≅11%; Baltic Sea: 84.8 mgC m-3≅9.7% of the mean biomass) when compared to the average values, MB seasonality is substantial (North Sea: 4.4 gC m-3≅222%; Baltic Sea: 1.86 gC m-3≅183% of the mean biomass)." Following a suggestion by reviewer 2 we also added a discussion on the MB seasonality to a revised version of the ms.

*P13L15: A minimum never falls, does it? A function of time falls until it reaches the minimum.* – Yes, right. changed to "reach"
*P13L30-31: Macrozoobenthos and fish can feed on their own excretions, since the energy contained in the carbon is not considered, correct? Also, there is no MB or fish respiration in the model, which would convert the organic carbon to DIC. So, there can be an infinite loop in the model where MB feeds and excretes, feeds and excretes, with no need to add energy from primary production. In Section 4 where you discuss future model improvements, could you comment on how to prevent this? I see, besides the way of explicitly defining "nutritional value classes" in detritus, the possibility to limit detritus consumption to ensure that at least a specific percentage of the total diet is fresh (non-detrital) material.*

> Daewel et al: We understand your concern, but we cannot agree on this

conclusion, and fear that the model description might be a bit unclear in that respect. On the one hand only parts of what is consumed can be assimilated while the rest can be considered as fecal excretion (this is probably what the reviewer is referring to). On the other hand the additional loss term for both fish and MB consists of a mortality and an excretion term. The mortality loss and fecal excretion contributes to the detritus pool, while the excretion loss is related to respiration of MB and fish and is considered directly as a source of dissolved inorganic nutrient ($PO_4$/$NH_4$ in principle also DIC if that were a state variable of the model)(see table 4 in the ms). Thus the feeding cycle features two specific loss term; i) the excretion term and ii) the remineralization of the dead organic matter, which would prevent the occurrence of such a self-containing feeding chain.

Action: We clarified these details in the model description of the ms (see also response to comment above P7L14). The role of the excretion term has been clarified. We also found an error in the last equation of table 4, which will be corrected.:" While fecal matter is accounted for through the use of assimilation efficiency, the excretion term from both fish and MB contributes directly to the nutrient reaction terms (see Table 4 equation for phosphate and ammonia)."

*P13L33: This is because the North Sea is a tidal sea, correct? I would consider it as helpful for the non-European readers to state this difference here.*

Daewel et al: Right, although the differences have been explained in the introduction, it is probably good to make that clear at this point.

Action: we changed the sentence to:" ... presumably due to the fact that a higher percentage of detritus is re-suspended in the tidal influenced, highly turbulent areas of the North Sea."

*P13L33: "Zooplankton and Phytoplankton is included" -> "Zooplankton and phytoplankton are included"?* –changed

*P15L5: "zooplankton form" -> "zooplankton forms"?* –changed

*P16L6-9: I would consider this as the main benefit of your model: You can obtain reasonable spatial patterns in zooplankton mortality without requiring data on planktivorous fish abundance. I would point this out already in the abstract and see it as the main point why your approach should be used.*

Daewel et al: Thanks for pointing that out. We will follow the suggestion and add this more prominent to the abstract of the revised ms:" By using this approach we addressed the above-mentioned closure term problem in lower trophic ecosystem modelling at very low computational costs and thus provide an efficient method that requires very few data to obtain spatially and temporally dynamic zooplankton mortality."

*P16L12-16: Couldn't you split your model fish into pelagic and demersal "feeding groups" based on their diet-dependent vertical distribution and then compare the spatial patterns?*

Daewel et al: We have tried to do so by differentiating between biomass in the water column and in the bottom layer (Suppl. Figure 2). However, clearly this is not directly comparable to the observation, where demersal and pelagic biomass is separated by species not actually by vertical appearance. Additionally the fish in our model has the chance to feed on all food sources at each time step. Nonetheless we agree that this comparison can be helpful and thus will include it in a revised version of the manuscript together with a discussion. One thing that becomes quite apparent is, as already suggested in the discussion, that the pelagic fish biomass is underrepresented by the model, since, in contrast to the observations, the bottom biomass exceeds that of the water column biomass. On the other hand, similar to the observation, both stocks contribute to the latitudinal gradient in biomass.

Note that there was a unit error in the original version of the Figure 10d that was corrected in the new figure (Suppl. Figure 2).
Action: We modified the original figure 10 by the Suppl. Figure 2 and changed the discussion accordingly.

*P16L25: What is zero at which boundary?* – Here we mean that no fish enters the area through the open boundaries to the North Atlantic by e.g. migration. Action: we entered the following half sentence: , meaning that no fish enters of leave the model area over the lateral boundaries,
*P17L8: "fish and MB is resolved" -> "are resolved"? -* changed
*P17L12: "for e.g." -> "e.g.", P18L31*: the same - changed

Reply reviewer 2

*Overall, this is a well written and interesting study linking a classic NPZD model to a single fish compartment. As a fisheries modeller and ecosystem modeller familiar with OSMOSE and Atlantic frameworks, I found the approach useful, however as the authors acknowledge, simplistic in its treatment of fish and fisheries. There is no mention of the impact that fisheries have on the ecosystem until the results section. There should be some introductory material about this as they are the biggest impact on fish populations, as many fisheries reduce more than half of fish biomass as a goal.*

Daewel et al: As our model is formulated, fisheries can only be targeted in a very simplistic way. In particular species specific fish stock information are not possible through our functional group type approach. That is the reason why fisheries has not been addressed more extensively in the ms. However, we agree with the reviewer that fishing indeed plays an important role for the fish biomass in both ecosystems, North Sea and Baltic Sea, and that given additional consideration to the topic might become an additional component of the model (see conclusion). That is why we will follow the reviewer's suggestion and add some relevant sentences to the introduction. "The two systems also differ substantially in terms of ecosystem dynamics. The North Sea is known as a highly productive area inhabited by more than 26 zooplankton taxa (Colebrook et al., 1984) and over 200 fish species (Daan et al., 1990), with highest biomasses distributed among demersal gadoids, flatfish, clupeids and sandeel (*Ammodytes marinus*) (Daan et al., 1990). Consequently the North Sea is economically highly relevant with nine nations fishing in the area with landings of currently about 2 million tons annually (ICES, 2018b). Compared to the North Sea, species composition in the Baltic Sea is primarily limited by the low salinities and encompasses only a few key players for zooplankton (Möllmann et al., 2000) and fish (Fennel, 2010). Thus, compared to the North Sea, commercial fishing in the Baltic Sea includes only a few stocks with total landings of over 0.6 million tons annually (ICES, 2018a). Both regions have in common that landings peaked in the 1970s and have substantially (ca 50%) decline since then. Thus fishing has a substantial impact on the overall fish biomass in the region."

*I would also like to see a more quantitative approach to calibrating the model to observed biomass of fish and fisheries catches. This data is spatially available, and a difference plot or map showing how well predicted vs observed fish biomasses compare in a spatially explicit analysis would be useful.*

Daewel et al: We have deliberately avoided an active calibration of the model. For the following reasons: i) The emerging pattern for fish from our model cannot fully represent the actual observed fish biomass due to the specific restriction applied to the functional group in our model (see new ms methods *page 7, lines 1-20*). Therefore a quantitative calibration to observed pattern would not necessarily improve the models parameterization. ii) Other than the reviewer implies, we do not think that the relevant data (spatial distribution of total fish biomass) actually are available, even if our model were to simulate the real overall fish distribution. However, we agree with the reviewer that a spatial comparison to fish distribution if desirable. Therefore we chose to estimate fish biomass from the IBTS surveys (Figure 10a-c), which is the only long-term, consistently sampled dataset on fish available in the region and compared those sampled distributions qualitatively to our estimated distribution (Fig. 10 & discussion in paragraph 3.3). Fisheries catch data are probably not applicable since fishery mortality is not explicitly parameterized in the model. It is quite possible that there are relevant datasets available that we are not aware of and we would be grateful if the reviewer would share this information with us.

*I'm also a bit concerned about the huge seasonality in biomass of the macrobenthos, much of which I presume is dominated by macroinvertebrates which don't vary in biomass as much as plankton communities do seasonally. & (page 13, line 11 - Is it reasonable for macrobenthos to vary that much annually? I would have expected there to be a much more constant standing stock similarly to fish. I'm thinking about the macro invertebrate community which doesn't vary that much through time. )*

Daewel et al: See also comment above. We think that the reviewer's concern is valid in the way that the one functional group for MB might not adequately address all processes responsible for MB variation. However a comparison to observed seasonality shows that the seasonality for MB can be quite high in both systems. We will thus add a discussion at the respective position in the manuscript.: "This is in line with observations on seasonality of benthic infauna at three different locations in the North Sea published by Reiss and Krönke (2005), who found maximum biomass in late summer. Although the observed seasonality showed the highest magnitude in the German Bight the seasonality was clear at all three locations. The authors concluded that of the potential relevant factors (food availability/quality, water temperature, predation, hydrodynamic stress) food

quality plays the major role for infauna seasonality, thus is strongly related to primary production. They also suggest food limitation and predation pressure to be the main processes for decrease in abundance during winter. The same authors also looked at seasonality in the epibenthic community (Reiss and Krönke, 2004) showing that the epifaunal biomass varies less seasonally, especially in the off shore region, and that the main processes causing seasonal variations are related to migratory behaviour, which is not covered by our model. For the Baltic Sea only very local studies in seasonality of MB are available of which some indicate locally strong seasonality (Anders and Möller, 1983), while in other regions no seasonal changes in biomass were observed due to the dominance of long-lived species (Persson, 1983). In general the comparison to observations indicates that on the one hand the model is able to represent the main seasonality in MB even though epi- and infauna are not separated. On the other hand in future study the consideration of an additional functional group encompassing longer-lived species will be required for addressing MB seasonality more correctly."

*It's not clear to me why the Baltic and North Seas were combined into one model, as they exhibit very different environmental and fish production regimes.*

Daewel et al: two reasons: First: Other then species specific models like OSMOSE and ATLANTIS, the methods we propose is based on a generic approach and shall be able to describe in a general way the transfer of matter and energy to the higher trophic levels. The specific advantage is, that it does not depend on species-specific information and the application to two systemically different regimes is in principle be possible and a good test to the method and assumptions made here. Second: Even though North Sea and Baltic Sea are very different systems both in physical as well as in biological characteristics, they are dynamically tightly coupled to each other. On the one hand the Baltic Sea dynamics strongly determine the conditions in the Norwegian trench and are thus relevant for simulating the northern North Sea dynamics. On the other hand, the coupling to the North Sea is essential to simulate timing and characteristic of the Major Baltic Inflow events and hence is relevant for the dynamics in the Baltic Sea. Since the model setup used here is computationally relatively cheap and runs quickly, there is no need to separate two systems that are closely interlinked.

*I think the closure terms where a lot of fish migration could be happening would be more important to focus on than connecting the two domains.*

Daewel et al: As we commented on in the conclusions (old version of the ms P19 L5 et sqq.) we are aware that the assumption of "no migration" is a shortcoming of the method and needs to be addressed in continuative studies. It is however beyond the scope of this ms.

*I would also like to see a more detailed treatment of fisheries mortality in the model, as this data is readily available and will be a huge driver of fish biomass given the very long exploitation history of the North Sea.*

Daewel et al: A better representation of fisheries mortality is desirable but requires developmental work for integrating a dynamical representation of fisheries mortality and is thus beyond the scope of this ms (also see conclusion old version of the ms P19 L20 et sqq.).

*Intro - You don't describe how this component contributes to the model and how the macrobenthos communities vary in the North and Baltic Seas.*

Daewel et al: We agree that this has not been sufficiently addressed in the introduction. Action: We add additional information on macrobenthos in the North Sea and Baltic Sea to the introduction section:" Studies on the food web dynamics of the North Sea and Baltic Sea highlight additionally the relevance of benthic fauna for fish consumption (Greenstreet et al., 1997; Tomczak et al., 2012). The term benthos refers generally to all organisms inhabiting the sea floor. A comprehensive review on the topic has been given by Kröncke and Bergfeld (2003). The faunal components encompass over 5000 species generally divided by size into microfauna, meiofauna, and macrofauna. Additional differentiation can be made under consideration of the vertical habitat structure, with infauna inhabiting the inner part of the sediment and epifauna living above the sediments. While in the North Sea the macrobenthos assemblages are basically structured based on the spatial distribution of sediment characteristics and depth, the Baltic sea community is additionally influenced by oxygen availability (Ekeroth et al., 2016) and salinity (Gogina et al., 2010). Besides its role as prey and predator in the marine foodweb, marcobenthos additionally influences nutrient effluxes from the sediments and thus can modify temporal and spatial patterns in nutrient concentrations (Ekeroth et al., 2016). „

*Results and Discussion - I suggest separating out your results first and then comparing to other studies. The way the two sections are intertwined makes it difficult to follow.*

Daewel et al: We understand the reviewer's opinion. However, it is quite common in complex modeling studies to combine the results and discussion section, to avoid a repetition of major results in a separate discussion section. We thought carefully about the structure and still think that a combined section is more appropriate for this study.

*page 2, lines 6-8 - the most common ecosystem models are Ecopath with Ecosim models which organize fish based on a combination of functional groups, species groups, and age-structured of species groups (see Tittensor et al. 2018 GMD).*

Daewel et al: Right! A short paragraph on EwE will be included in the introduction." One of the most commonly used food-web modelling tools is the Ecopath with Ecosim (EwE) modelling software (Christensen and Walters, 2004), which provides an instantaneous snapshot of the trophic mass balance in marine food webs. Together with the dynamical modelling capability (Ecosim) and a tool that replicates the model on a spatial grid (Ecospace) it allows 2d estimates of the systems response to e.g. policy measures. However the approach still falls short in simulating ecosystem dynamics on high temporal and 3d spatial resolution."

*page 3, lines 16-18 - How can questions about food webs be tested when there is only one fish functional group? There are different trophic levels of fish that are harvested which exert different controls on the macro food web. For example, forage fish have been shown to be important prey for many higher trophic levels and their exploitation has different effects than harvesting top predators (see Smith et al. 2010 Science).*

Daewel et al: We agree that these kind of food-web interactions cannot be addressed by the current formulation of the model. However, the model presented in the ms is a first approach that can be further developed to represent a more complex food web (by e.g. distributing the fish group into separate feeding guilds), which can then be used to address this kind of question. We will make this more clearly in a revised version of the manuscript. Added to the introduction: ".The approach we use cannot address changes in ecosystem structure related to variations in the fish assemblage or selected fishing activities. It does, however, provide the potential for further developments towards a more complex food web (e.g. by distributing fish into separate feeding guilds), which will then allow us to address specific changes in food web structure. "

*page 4, line 1 - Awkward start to the sentence, suggest restructuring.*

Daewel et al: Will be changed to: "Here we present a functional-type, E2E modelling approach, which relates food availability to potential fish growth and biomass distributions. In this manuscript we introduce..."

*page 4, lines 16-17 - Awkward sentence, suggest combining with previous sentence in parentheses. -*
changed.

*page 6, line 25 - What are the constraints for vertical fish movement based on oxygen and temperature limitations? This is one constraint of using only one fish group as there is a lot of variability among fish species in sensitivity to environmental conditions.*

Daewel et al: Fish consumption is constrained by oxygen availability such the fish would not migrate into low oxygen regions. This information was missing in the ms and will be added to the method section. Temperature only plays a role for metabolic rates. : "Note that, since fish do not tolerate anoxic conditions, only grid cells featuring positive oxygen concentrations were considered for the estimate of fish consumption. „
The advantage of the generic functional group approach used in the model is that we simplify a complex community structure (also for phytoplankton and zooplankton) and reduce the information to the basic common features thus avoiding a huge parameter set and data requirements. Still the model is able to simulate relevant ecosystem dynamics.
We will add an additional explanation to the manuscript for clarification.
Action: the following sentence has been added to the conclusion:" The advantage of the generic functional group approach used in the model for all trophic levels is that we can simplify a complex community structure and reduce the information to the basic common features thus avoiding a huge parameter set and data requirements. Still the model is able to simulate relevant ecosystem dynamics on high spatial and temporal resolutions with relatively low computational requirements."

*page 7, lines 17-18 - This sentence can be combined with the previous sentence.* –changed

*page 8, lines 1-12 - What about fisheries mortality on the fish compartment? Was this not included in the model? For the North Sea, this would comprise a significant proportion of total mortality.*

> Daewel et al: Fisheries has not been included in the reference simulation where mortality accounts for natural mortality and predation losses, which allows us to estimate the undisturbed "fish production potential". We will clarify this in a revised version of the manuscript. To understand and clarify the impact of additional fisheries mortality we have additionally included the "fisheries scenarios" in the ms.
> Action: Added to the method description: "Fisheries mortality has not ben considered for the standard simulation, but was explicitly addressed in additional scenario experiments as described in section 2.3."

*page 10, lines 11-16 - This is the first mention of fisheries, if they are a component of the model, then they should have been included much earlier in the manuscript. How was the loss rate calculated? Much greater detail about the fisheries data that was used and how this was applied need to be included.*

> Daewel et al: Since simulating and understanding fisheries impacts on the ecosystem is not (and cannot be, due to the model simplifications applied) the primary research question addressed by the model, fisheries has not been addressed in the reference simulation (see comment above).
> Therefore we included the scenarios to understand the effect of fisheries on the results. From estimated biomass and reported fisheries landings (see below our response to *page 12, line 28 - page 13, line 9*) we see that biomass losses due to fisheries are in both regions in the range of 20%-50% of the fish biomass per year. For convenience we decided to receive the fisheries mortality rate by scaling the natural mortality rate ($=0.001d^{-1} =0.365yr^{-1}$) with 0.5, 1 & 2 . Thus receiving mortality rates that are representative for the bandwidth of the observed fisheries loss rates.
> We will add a related explanation to a revised version of the manuscript.
> Action: the following sentence has been added to the scenario description: "Following the fisheries overview of the North Sea and Baltic Sea region as published by ICES (2018a, 2018b) biomass losses due to fisheries are in both regions in the range of 20%-50% of the total fish biomass per year. For convenience we therefore decided to receive the fisheries mortality rate by scaling the natural mortality rate (=0.001d-1 =0.365yr-1) with 0.5, 1 & 2."

*page 10, lines 18-23 - How was this value of 20% less loss rate derived? Any empirical data that supports this value?*

> Daewel at al: See explanation in the methods section (P7 and paragraph below)

*page 12, lines 24-26 - This is a reason why it is important to consider functional groups/species/age classes/size classes when including fish in ecosystem models.*

> Daewel et al: As in many other models this depends on the research question that should be addressed with the model. If the main aim is indeed to estimate species-specific stock biomass or fisheries landings, we agree that the model requires a much greater detail then what we presented here. On the other hand if the question is related to the overall productivity of the ecosystem, energy and matter transfer processes and long-term variation of the latter, which is aimed for here, a more simplified approach with less data requirements and low computational costs is a helpful method. Of course that causes difficulties when comparing to observations. However, all models (also those that actually resolve species/age/size) are simplifications in a way and a large number of groups comes with a high degree of uncertainty in parameter settings and large computational costs.

*page 12, line 28 - page 13, line 9 - How do the fish biomass estimates compare to reported fisheries landings from the regions? Is there enough biomass to support known landings?*

> Daewel et al: Following ICES, fisheries during the 1980's were in the range of 0.7-1 million tons in the Baltic Sea and 2-3 million tons in North Sea. Despite that the model underestimates fish production due to the no horizontal migration assumption and no fish migration over the lateral boundaries, the models' estimates of fish biomass in the North Sea: 5.13-10.27 million tons and in the Baltic Sea: 3.47-6.93 million tons would support the fisheries landing during that time. We will add explanations in the revised manuscript.

*page 14, line 13 - You haven't introduced this analysis in the methods section to describe what it is.*

> Daewel et al: We understand the need for a more detailed description of the method. Since the method has been in detail introduced in an earlier manuscript, we suggest adding the method description to the ms as follows: The EOF analysis is a statistical method to understand major

modes of variability in multidimensional data fields. A detailed description on how this method has been applied is given in Daewel et al. (2015): "The annual values of the spatially explicit variable field form a NxM matrix $\chi$ (N: number of years; M: number of wet grid points). The empirical modes are given by the K eigenvectors of the covariance matrix with non-zero eigenvalues. Those modes are temporally constant and have the spatially variable pattern pk(m=1,…,M) where k=1,…,K. The time evolution Ak(t=1,…,N) of each mode can then be obtained by projecting pk(m) onto the original data field $\chi$ such that $\chi(t, m) = \sum_{k=1}^{K} p_k(m)A_k(t)$. In the following we will refer to Ak(t) as the principal components (PC) and to pk(m) as empirical orthogonal function (EOF). The percentage of the variance of the field $\chi$ explained by mode k is determined by the respective eigenvalues and is referred to as the global explained variance ηg(k). Before using the method to analyse the spatiotemporal dynamics of the field, the data were demeaned (to account for the variability only) and normalized (to allow an analysis of the variability independent of its amplitude). The identified modes are not necessarily equally significant in all grid points of the data field. Thus, the local explained variance ηlocal,k(m) could provide additional information about the regional relevance of an EOF mode and the corresponding PC in percent:

$$\eta_{local}{}^k(m) = \left[ 1 - \frac{Var\big(\chi(m,t)-p^k(m)A^k(t)\big)}{Var(\chi(m,t))} \right] \cdot 100 \ ,$$

$$(12)$$

[revised manuscript text omitted]

Ute Daewel 13.3.2019 15:25

Ute Daewel 15.2.2019 11:20

Ute Daewel 15.2.2019 11:23

Ute Daewel 15.2.2019 12:53

Ute Daewel 15.2.2019 10:35

Ute Daewel 15.2.2019 10:35

Ute Daewel 15.2.2019 10:36

Ute Daewel 15.2.2019 10:51

Ute Daewel 4.3.2019 17:11

Ute Daewel 4.3.2019 16:54
Formatted

Ute Daewel 4.3.2019 16:54
Formatted

Ute Daewel 4.3.2019 16:54
Formatted

Ute Daewel 4.3.2019 16:54
Formatted

Ute Daewel 4.3.2019 16:54
Formatted

Ute Daewel 4.3.2019 16:54
Formatted

Ute Daewel 4.3.2019 16:54
Formatted

Ute Daewel 4.3.2019 16:54
Formatted

Ute Daewel 4.3.2019 16:54
Formatted

Ute Daewel 4.3.2019 16:54
Formatted

Ute Daewel 4.3.2019 16:54
Formatted

Ute Daewel 4.3.2019 17:11

[revised manuscript text omitted]

Ute Daewel 5.3.2019 10:33
Ute Daewel 5.3.2019 10:33
Ute Daewel 5.3.2019 11:33
Ute Daewel 13.3.2019 15:37
Ute Daewel 5.3.2019 11:35
Ute Daewel 5.3.2019 16:14
Ute Daewel 5.3.2019 16:15
Ute Daewel 13.3.2019 15:37
Ute Daewel 5.3.2019 16:15
Ute Daewel 5.3.2019 16:16
Ute Daewel 5.3.2019 16:19
Formatted ... [10]
Ute Daewel 5.3.2019 16:19
Formatted ... [11]
Ute Daewel 5.3.2019 16:23
Formatted ... [12]
Unknown
Field Code Changed ... [13]
Ute Daewel 5.3.2019 16:25
Ute Daewel 5.3.2019 16:26
Ute Daewel 5.3.2019 16:32
Formatted ... [14]
Ute Daewel 5.3.2019 16:32
Formatted ... [15]
Ute Daewel 13.3.2019 15:50
Formatted ... [16]
Ute Daewel 5.3.2019 16:35
Ute Daewel 13.3.2019 15:51
Unknown
Field Code Changed ... [17]
Ute Daewel 13.3.2019 15:52

[revised manuscript text omitted]